# Modulating Pt-O-Pt atomic clusters with isolated cobalt atoms for enhanced hydrogen evolution catalysis

Yufei Zhao [1], Priyank V. Kumar[1], Xin Tan[2], Xinxin Lu[3], Xiaofeng Zhu[1], Junjie Jiang[1], Jian Pan [1], Shibo Xi[4], Hui Ying Yang [5], Zhipeng Ma[1], Tao Wan[6], Dewei Chu [6], Wenjie Jiang[1], Sean C. Smith[2], Rose Amal [1], Zhaojun Han [1,7✉] & Xunyu Lu [1✉]

Platinum is the most efficient catalyst for hydrogen evolution reaction in acidic conditions, but its widespread use has been impeded by scarcity and high cost. Herein, Pt atomic clusters (Pt ACs) containing Pt-O-Pt units were prepared using Co/N co-doped carbon (CoNC) as support. Pt ACs are anchored to single Co atoms on CoNC by forming strong interactions. Pt-ACs/CoNC exhibits only 24 mV overpotential at 10 mA cm$^{-2}$ and a high mass activity of 28.6 A mg$^{-1}$ at 50 mV, which is more than 6 times higher than commercial Pt/C with any Pt loadings. Spectroscopic measurements and computational modeling reveal the enhanced hydrogen generation activity attributes to the charge redistribution between Pt and O atoms in Pt-O-Pt units, making Pt atoms the main active sites and O linkers the assistants, thus optimizing the proton adsorption and hydrogen desorption. This work opens an avenue to fabricate noble-metal-based ACs stabilized by single-atom catalysts with desired properties for electrocatalysis.

[1] Particles and Catalysis Research Laboratory, School of Chemical Engineering, The University of New South Wales, Sydney, NSW 2052, Australia. [2] Integrated Materials Design Laboratory, Department of Materials Physics, Research School of Physics, Australian National University, Canberra, ATC 2601, Australia. [3] School of Science and Engineering, The Chinese University of Hong Kong (Shenzhen), Shenzhen 518172, China. [4] Institute of Chemical & Engineering Sciences, Agency for Science, Technology and Research (A*STAR), Singapore 627833, Singapore. [5] Singapore University of Technology and Design, 8 Somapah road, Singapore 487372, Singapore. [6] School of Materials Science and Engineering, The University of New South Wales, Sydney, NSW 2052, Australia. [7] CSIRO Manufacturing, 36 Bradfield Road, Lindfield, NSW 2070, Australia. ✉email: zhaojun.han@unsw.edu.au; xunyu.lu@unsw.edu.au

Hydrogen ($H_2$) is a promising energy carrier owing to its high mass energy density and cleanness. The traditional way to produce hydrogen, namely the steam reforming of natural gas, is not sustainable as it requires fossil fuels (e.g., natural gas) as the feedstock meanwhile generates a tremendous amount of carbon dioxide. Electrochemical water splitting powered by renewable electricity is an attractive approach to generate hydrogen with zero emissions[1–4]. Moreover, by storing the excessive electrical energy in the form of chemical energy, this technology can also effectively overcome the intermittency issue of renewable energy resources, such as solar energy[5]. Proton exchange membrane (PEM) electrolyzer is a popular type of device for hydrogen generation through water splitting. PEM electrolyzer is simple and compact, which is capable of generating high purity hydrogen at relatively high current densities[6,7]. So far, the most efficient catalysts for hydrogen evolution reaction (HER) under the acidic environment within a PEM electrolyzer are still based on Pt, while its scarcity and high cost have posed unfavorable restraints for the large-scale rollout of this device to produce hydrogen[8]. Therefore, strategies that can significantly lower the amount of Pt without sacrificing the electrocatalytic performances are highly sought after to facilitate the widespread deployment of PEM electrolyzers.

Single-atom catalysts (SACs) are a type of emerging catalysts for electrocatalysis. The atomically dispersed properties of metal sites in SACs make each of them accessible and catalytically active in electrocatalytic processes, thereby lowering the amount of metal species required as well as the corresponding cost meanwhile maintaining the high catalytic performance[9–13]. Some SACs, such as $Pt_1$/N-C, $Pt_{SA}$/$WO_{3-x}$ and $Ru_{SA}$-N-S-$Ti_3C_2T_x$, have exhibited outstanding HER catalytic capabilities in acidic media[14–16]. However, metal atoms are sparsely distributed in SACs, which has rendered them far apart from each other. As a consequence, the interactions between these metal atoms are generally negligible and are not able to influence the electrocatalytic processes. Further modulation of the metal sites within SACs at the atomic level by introducing neighbouring metal atoms may provide an opportunity to further boost their intrinsic catalytic activity. Recently, atomic clusters (ACs) composed of several surface metal atoms have attracted increased research interest[17–20]. Metal ACs can offer the active sites with adjacent metal atoms while maintaining them as individual sites thereby maximizing the metal utilization efficiency. The potential synergistic interactions between the metal atoms result in unique electronic and geometric properties of the ACs that are highly beneficial for electrocatalysis[21]. It is demonstrated that Pt ACs that are supported on $TiO_2$ and defective graphene are appealing catalysts for HER[22,23]. However, constructing stable ACs on conductive supports for electrocatalysis remains a great challenge due to their high surface energy. The sub-nanometric ACs tend to aggregate into nanoparticles, thereby losing their unique properties as well as the promising catalytic performance. To stabilize the metal ACs on the supports, dedicated and sophisticated synthetic procedures are normally required, such as precursor-preselected strategy, host-guest strategy and atomic layer deposition method[20,24–27], which are not suitable for large-scale applications. Thus, a facile and efficient method to fabricate stable ACs with strong metal-substrate interactions is highly desired.

In this work, isolated Co atoms and N co-doped porous carbon (CoNC) is employed as a unique substrate to direct the formation of Pt ACs (Pt-ACs/CoNC). The Co atoms in CoNC act as the anchoring sites for the Pt species, forming strong metal-support interactions that can effectively prevent the unwanted aggregation of Pt atoms into nanoparticles. X-ray absorption spectroscopy (XAS) and density functional theory (DFT) have revealed the Pt ACs are comprised of Pt atoms bridging with O atoms, forming Pt-O-Pt unit with a lower Pt oxidation state compared to the Pt atoms in SACs, attributing to the less charge delocalized from Pt to the surrounding O atoms in Pt-ACs/CoNC. The presence of neighbouring and electron-enriched Pt atoms in Pt-O-Pt unit makes them function as the main active sites for $H_2$ production, and these O linkers further facilitate adsorbed H* transfer and $H_2$ desorption. The resultant Pt-ACs/CoNC exhibits an exceptional catalytic activity towards HER, requiring merely 24 mV of overpotential to achieve a current density of 10 mA cm$^{-2}$. Moreover, the mass activity of Pt-ACs/CoNC is 28.6 A mg$^{-1}$ at the overpotential of 50 mV, which is superior to the commercial Pt/C with 20 wt% of Pt under identical testing conditions. This simple method has been extended to prepare Ru ACs and Ir ACs that are supported on CoNC, showing desirable properties for electrocatalysis.

## Results

**The preparation and characterization of Pt-ACs/CoNC.** The Pt-ACs/CoNC composite is synthesized by anchoring the atomic-layered Pt ACs on a highly porous CoNC substrate via a simple wet-impregnation process (Fig. 1a). Firstly, the CoNC substrate is prepared by carbonizing the mixture of *o*-phenylenediamine, cobalt nitrate and silica nanosphere templates (Experimental details refer to Methods)[28]. The high porosity of CoNC has resulted in a large specific surface area of 1147.4 m$^2$ g$^{-1}$ (Supplementary Fig. 1), which is beneficial for the electron transfer and access to reactants during HER. The isolated Co atoms on the CoNC substrate with a mass loading of 3.5 wt% have provided abundant anchoring sites for the formation of Pt ACs (STEM images and mapping results of CoNC in Supplementary Fig. 2). X-ray photoelectron spectroscopy (XPS) and XAS analyses in Supplementary Fig. 3a–d reveal the isolated Co atoms are positively charged and stabilized by four surrounding N atoms in the equatorial plane and one O atom in the axial position. These O atoms above Co single atoms can capture the incorporated Pt ions to form the protrusive atomic structure (Supplementary Fig. 3e). As a result, after adding a certain amount of Pt precursors followed by a subsequent low-temperature treatment, the stable Pt ACs are formed on the surface of CoNC substrate.

The as-prepared Pt-ACs/CoNC was characterized by the high angle annular dark-field scanning transmission electron microscopy (HAADF-STEM). As shown in Fig. 1b and Supplementary Fig. 4, the Pt-ACs/CoNC composite has inherited the porous structure of CoNC and no metal nanoparticles (NPs) can be spotted. Instead, numerous clusters with an average size of less than 1 nm are found homogeneously dispersed on the carbon support. The loading of Pt was determined to be 0.52 wt% by the inductively coupled plasma-mass spectrometry (ICP-MS). Intriguingly, the Pt clusters are distributed in ensembles that Pt atoms locate next to each other roughly on the same plane (Fig. 1c), which is entirely different from the structure of a nanoparticle. The magnified image of the area marked in Fig. 1c reveals that Co atoms are scattered around Pt ACs (dots with different contrasts, Fig. 1d), indicating the possible interactions between Pt clusters and Co SAs. We further performed STEM simulation and DFT optimization (details refer to the DFT calculation sections in Methods) on a Pt AC supported by Co SAs, which is consistent with the experimental STEM image (Fig. 1d–f). Figure 1h represents the extracted line profiles from Fig. 1g. The Pt atoms show similar height with one-atomic-layer thickness and the close dispersion of Co and Pt atoms is also observed (two atoms with different heights because of the different atomic radii). The extracted line profiles demonstrate the atomically dispersed properties of the Pt atom and the existence of possible interactions between Pt and Co species[27]. Energy-dispersive X-

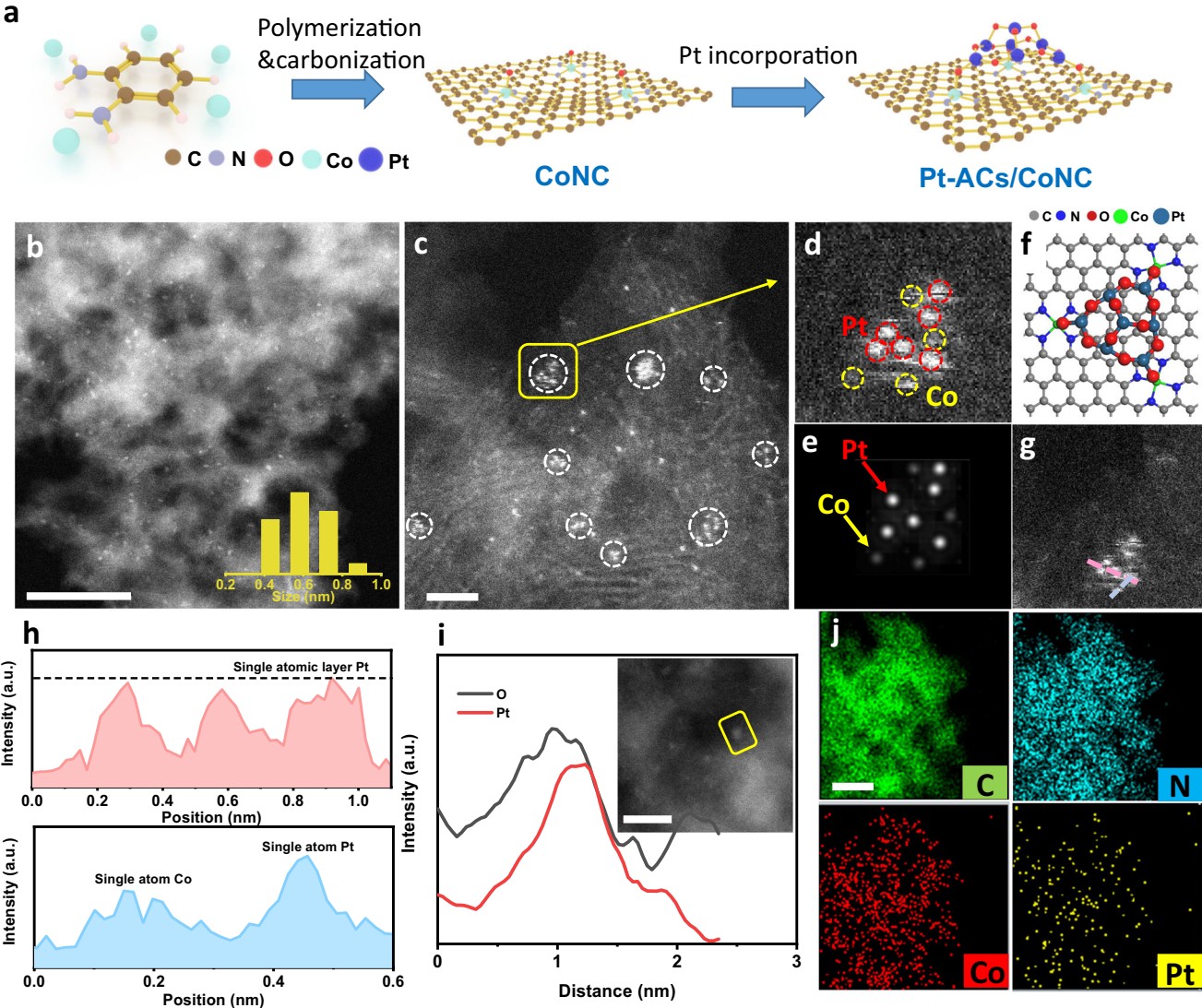

**Fig. 1 Structural characterization of Pt-ACs/CoNC. a** Schematic illustration of the preparation of Pt-ACs/CoNC. **b**, **c** STEM images of Pt-ACs/CoNC. Scale bar, 20 and 2 nm, respectively. **d**–**f** Magnified HAADF-STEM image of Pt-ACs/CoNC, its corresponding simulated image and DFT optimized structure. **g**–**h** The extracted line profiles along with the red and blue directions. **i** EDS analysis of Pt-ACs/CoNC, and the inset is the HAADF-STEM image at a low voltage of 60 kV. Scale bar, 5 nm. **j** STEM-EDS elemental mapping of Pt-ACs/CoNC. Scale bar, 10 nm.

ray spectroscopy (EDS) line scan results regarding to two ACs as in Fig. 1i and Supplementary Fig. 5 reveal the similar distribution trends of both O and Pt over the clusters, which indicates that O atoms may function as linkers to connect the Pt atoms in Pt ACs[29]. EDS mapping results in Fig. 1j and Supplementary Fig. 6 show the existence of C, N, O, Co, and Pt elements, as well as their homogeneous distribution over the entire area measured.

By varying the amount of Pt precursors in the synthetic procedure (see details in Methods), Pt single atoms (SAs) and nanoparticles (NPs) can be successfully obtained on the CoNC substrate with loadings of 0.17 and 1.47 wt%, as measured by ICP-MS, respectively. The HAADF-STEM images of Pt-SAs/CoNC (Supplementary Fig. 7a, b) show the absence of both nanoparticles and atomic clusters. Instead, the presence of numerous bright dots with different contrast is spotted. The observed dots with different contrast indicate the existence of two distinctive metal elements (in our case are Pt and Co atoms), together with other non-metal elements such as C, N and O (Supplementary Fig. 7c). Furthermore, the Pt and Co atoms are in close proximity to each other (as circled in Supplementary

Fig. 7b), suggesting the formation of possible interactions between them. By contrast, numerous Pt NPs can be clearly observed in the Pt-NPs/CoNC composite (Supplementary Fig. 8), with a size ranging from 1 to 3 nm. Collectively, the above results indicate that when Pt ions are added into CoNC, they tend to initially attach to the O atoms above the Co SAs, forming the Pt SAs protrusive structure. This structure makes it easier for further attachment of Pt atoms to form Pt ACs supported by the isolated Co atoms. Further increase of the Pt content will eventually result in the formation of Pt NPs.

The X-ray diffraction (XRD) patterns of CoNC, Pt-SAs/CoNC, Pt-ACs/CoNC and Pt-NPs/CoNC are shown in Supplementary Fig. 9a. Only two dominant peaks at around 26° and 44° can be observed in all the patterns, ascribing to the (002) and (101) planes of the graphitic carbon. No metal-related peaks are observed in the XRD patterns, including the Pt-NPs/CoNC composite, indicating the ultra-small sizes of the metal species in these samples render them undetectable by XRD[30]. Supplementary Fig. 9b represents the Raman spectra of CoNC, Pt-SAs/CoNC, Pt-ACs/CoNC and Pt-NPs/CoNC. The spectra show a

typical D band at ~1338 cm$^{-1}$ and the G band at ~1590 cm$^{-1}$, which can be ascribed to the carbon lattice defects and sp$^2$-hybridized carbon atoms in the CoNC substrate[31]. All samples prepared herein exhibit similar $I_D/I_G$ values (within a small range of 1.14–1.17), indicating the incorporation of Pt species has a negligible effect on the defective or graphitic properties of the CoNC matrix. Supplementary Fig. 10 shows the nitrogen adsorption/desorption isotherms of Pt-SAs/CoNC, Pt-ACs/CoNC and Pt-NPs/CoNC composites and their corresponding Brunauer-Emmett-Teller (BET) surface areas derived from these isothermals. The BET surface areas decrease slightly with an increase in the Pt content. Nevertheless, owing to the ultralow loading of Pt species, the Pt-ACs/CoNC composite manages to maintain a high BET surface area (1070 m$^2$ g$^{-1}$), which is highly beneficial to HER catalysis.

**The structure identification of Pt-ACs/CoNC.** The electronic structure and coordination environment of Pt-ACs/CoNC were studied by XPS and XAS measurements. The high-resolution XPS spectra of N 1s for Pt-SAs/CoNC, Pt-ACs/CoNC and Pt-NPs/CoNC in Supplementary Fig. 11a show similar peak positions and intensities to that of CoNC, which can be deconvoluted into three peaks at 398.64, 401.00, 403.00 eV, corresponding to pyridinic-N/metal-N, graphitic N, and oxidized N, respectively[32,33]. Similarly, O 1s spectra of CoNC, Pt-SAs/CoNC, Pt-ACs/CoNC and Pt-NPs/CoNC in Supplementary Fig. 11b all exhibit peaks at the positions around 532.9 and 531.1 eV, attributing to OH$^-$ and O$^{2-}$, respectively[34]. The intensities of these peaks, however, change significantly when varying the Pt loading. The intensity of the peak belonging to O$^{2-}$ increases with higher Pt loading (from SAs to NPs), whereas the intensity of the OH$^-$ peak decreases correspondingly, attributing to the formed Pt-O bonds when Pt ions are incorporated. The high-resolution Co 2p spectrum of CoNC in Fig. 2a shows the peaks at 779.92 and 795.18 eV, revealing that Co species are at their oxidation states, further demonstrating their isolated properties as SAs[9]. Intriguingly, slight shifts of these peaks are spotted in Pt-SAs/CoNC and Pt-ACs/CoNC (more obvious), suggesting that the Pt species interacted with Co SAs on the carbon substrate, which demonstrate the incorporation of atomically dispersed Pt species to CoNC exhibits an obvious electronic modulation of Co atoms and establishes a strong metal-support interaction. This peak shift for Co 2p spectrum is not obvious when Pt NPs are formed in the composite, attributing to the Pt NPs that have concealed the signal of the majority interacted Co species. The XPS spectra of Pt 4f in Supplementary Fig. 11c show the Pt species possess cationic nature in both Pt-SAs/CoNC and Pt-ACs/CoNC composites, while the Pt species in Pt-NPs/CoNC are partially in the metallic form. Moreover, the valence state of Pt in Pt-ACs/CoNC is lower than that in Pt-SAs/CoNC, evidenced by the negatively shifted of Pt 4f peaks (Supplementary Fig. 11c), making these clusters more efficient for the reduction reaction[14].

The X-ray absorption near edge structure (XANES) results of Co K-edge in Supplementary Fig. 12a show that the absorption edges of CoNC, Pt-SAs/CoNC, Pt-ACs/CoNC and Pt-NPs/CoNC locate between Co foil and Co$_3$O$_4$, indicating that the valence states of Co species in all samples are lower than 2+. Moreover, the slightly different white-line intensities in Co K-edge after Pt incorporation indicate the slight alternation in charge density around Co atoms, and this trend is similar to the XPS results. The normalized XANES results of Pt L-edge of Pt-SAs/CoNC, Pt-ACs/CoNC and Pt-NPs/CoNC samples show the white-line intensities between Pt foil and PtO$_2$ (Fig. 2b), indicating the Pt species in all three catalysts are positively charged. The white-line intensity decreased from Pt-SAs/CoNC to Pt-ACs/CoNC

(consistent with XPS results in Supplementary Fig. 11c), verifying a relatively lower valence state of Pt in Pt-ACs/CoNC. To acquire an accurate estimation, we calculate the valence state of Pt in these three samples based on the Pt foil and PtO$_2$. As shown in Fig. 2c and Supplementary Fig. 13, Pt-SAs/CoNC displays a higher valence state (1.76) than Pt-ACs/CoNC (1.59). The lower valence state of Pt-ACs/CoNC implies increased d-band electrons of Pt species, which is deemed to be highly beneficial to HER[14]. The coordination environment has been investigated by extended X-ray absorption fine structure (EXAFS) results. In Fig. 2d, the EXAFS of Co R space in CoNC presents a distinguished peak located at around 1.54 Å, attributed to the Co-N/O scattering path. The Co R spaces of Pt-SAs/CoNC, Pt-ACs/CoNC and Pt-NPs/CoNC all exhibit similar peak positions to that of CoNC, indicating Co atoms in these catalysts share an analogous coordination environment to the bare CoNC. In Figs. 2e, f, the absence of Pt-Pt contribution at ~2.5 Å in the EXAFS results of Pt L$_3$ edge in Pt-SAs/CoNC and Pt-ACs/CoNC suggests no Pt nanoparticles or Pt-Pt interactions are present in these two catalysts, whereas the Pt-Pt peaks emerge in Pt-NPs/CoNC. Instead, the only dominant peak for both Pt-SAs/CoNC and Pt-ACs/CoNC locates around 1.65 Å, sharing a similar position to the typical exclusive Pt-O coordination environment in PtO$_2$, which demonstrates the Pt atoms in Pt-SAs/CoNC and Pt-ACs/CoNC are coordinated by O atoms. The wavelet transform (WT) of the $k^3$-weighted EXAFS spectra was shown in Fig. 2f, which can directly reflect the structure information in both $k$ and R spaces. The results indicate the WT contour plots of Pt-SAs/CoNC and Pt-ACs/CoNC display only one intensity at around 5 Å, attributed to the Pt-light atoms bonding, which further confirms the atomically dispersed states of Pt atoms in both Pt-SAs/CoNC and Pt-ACs/CoNC[35,36]. The fitting results of Pt-SAs/CoNC (Supplementary Fig. 14) show one O atom between Co and Pt atoms, and one chemisorbed O$_2$ molecule at the terminal position. In contrast, the fitting results of Pt-ACs/CoNC (Fig. 2e) indicate the Pt atoms are separated by O atoms to form Pt-O-Pt units and each Pt atom is stabilized by three O atoms through Pt-O bonds in Pt-ACs/CoNC (inset of Fig. 2e), which well aligns with the EXAFS results.

It is well established that the overall property of Pt-ACs/CoNC can be affected significantly by both Pt ACs and CoNC. The CoNC matrix with atomically dispersed Co SAs plays an essential role in the immobilization of Pt ACs. The Pt clusters can be stabilized by the Co SAs through O atoms by forming Co-O-Pt bonds at the interface. The direct interactions between CoN$_4$ structure and Pt ACs allow the electrons to transfer between Co SAs and Pt ACs (XPS and XAS results in Fig. 2, Supplementary Figs. 11 and 12), restricting the unwanted migration of Pt ACs during synthesis and the subsequent catalytic processes. A comparison experiment was conducted by immobilizing Pt species directly on N-doped carbon (NC) without the presence of isolated Co atoms. As shown in Supplementary Fig. 15, numerous Pt NPs, rather than Pt ACs, are observed on the NC matrix with merely 0.5 wt% of Pt, further confirming the crucial role of Co SAs played in directing the formation of Pt ACs.

**Electrocatalytic HER performance of Pt-ACs/CoNC.** The HER catalytic activity of the as-prepared samples was evaluated by a three-electrode setup in 0.5 M H$_2$SO$_4$ solution at room temperature (see details in the Methods). The working electrodes are prepared by drop-casting the catalyst inks onto the surface of the rotating disk electrode (RDE). As shown in Fig. 3a, CoNC shows a relatively poor HER activity with a high onset potential of 131 mV and an overpotential of 256 mV to achieve the current density ($j$) of 10 mA cm$^{-2}$. In contrast, Pt-SAs/CoNC, Pt-ACs/

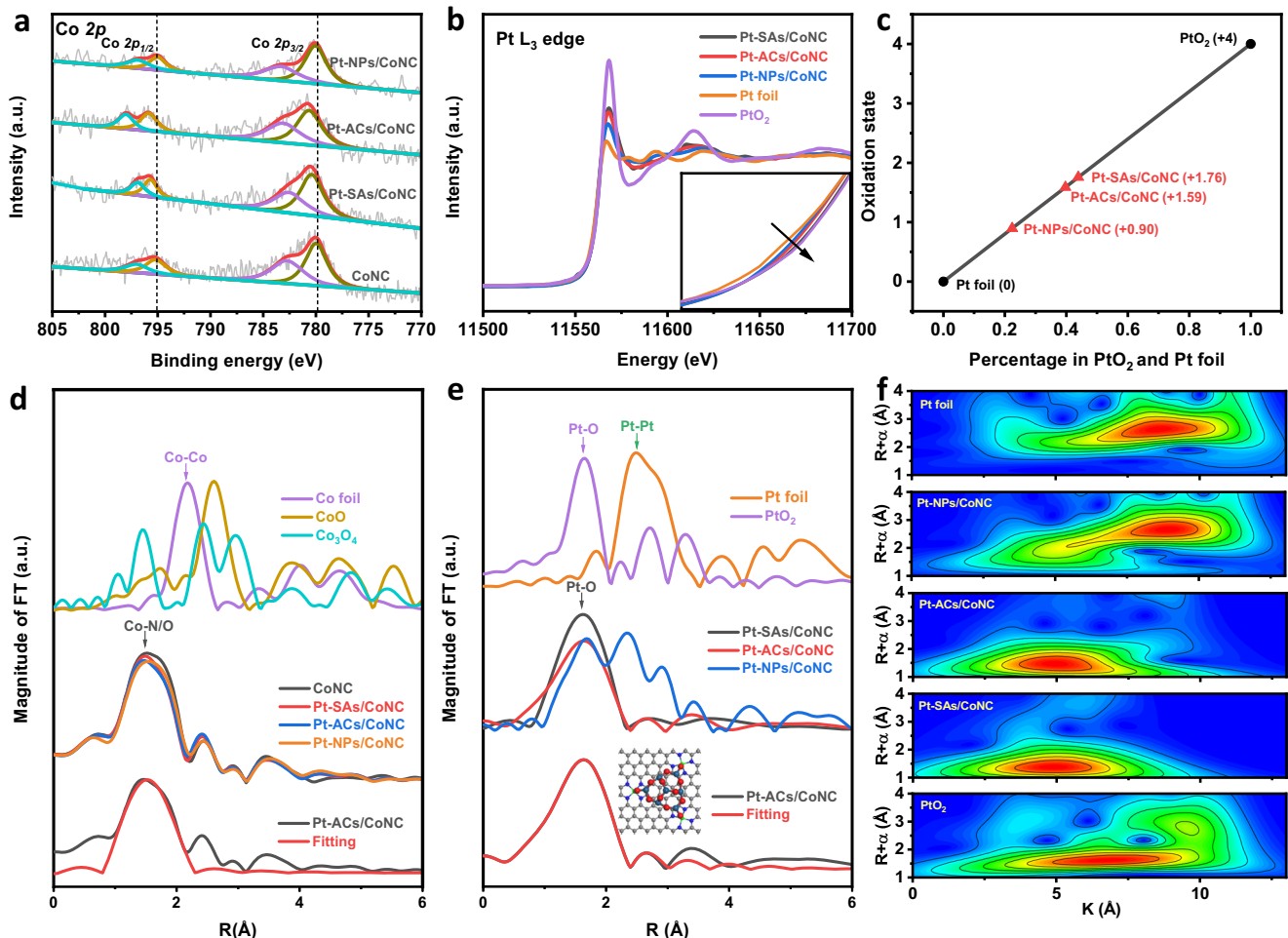

**Fig. 2 Structural characterizations of Pt-ACs/CoNC. a** The high-resolution XPS spectra of Co $2p$ in CoNC, Pt-SAs/CoNC, Pt-ACs/CoNC and Pt-NPs/CoNC. **b** Normalized XANES spectra at the Pt $L_3$-edge of Pt foil, $PtO_2$ and Pt-based catalysts. **c** The fitted average oxidation states of Pt in Pt-SAs/CoNC, Pt-ACs/CoNC and Pt-NPs/CoNC. **d** The normalized XANES spectra and the $k^3$-weighted Fourier transform of EXAFS spectra at Co K-edge of CoNC, Pt-SAs/CoNC, Pt-ACs/CoNC, Pt-NPs/CoNC and the reference materials. EXAFS curves between the experimental data and the fit of Pt-ACs/CoNC at Co K-edge. **e** The $k^3$-weighted Fourier transform of EXAFS spectra derived from EXAFS of Pt foil, $PtO_2$, Pt-SAs/CoNC, Pt-ACs/CoNC and Pt-NPs/CoNC. EXAFS curves between the experimental data and the fit of Pt-ACs/CoNC, the inset is the fitted structure. **f** Wavelet transform for the $k^3$-weighted EXAFS spectra of Pt foil, $PtO_2$, Pt-SAs/CoNC, Pt-ACs/CoNC and Pt-NPs/CoNC.

CoNC and Pt-NPs/CoNC exhibit significantly enhanced HER catalytic activity with near-zero onset potential, indicating that the Pt-based catalysts are much more favourable for HER (Supplementary Fig. 16a). Pt-SAs/CoNC delivers a $j$ of 10 mA cm$^{-2}$ at an overpotential of 57 mV. This performance is comparable to other Pt-based SACs under similar testing conditions (Supplementary Table 2). When Pt-ACs/CoNC is used as the catalyst, the overpotential for achieving a $j$ of 10 mA cm$^{-2}$ is reduced to merely 24 mV, which is slightly superior to that achieved with the commercial Pt/C (same total loading on the working electrode, Supplementary Fig. 16b) with a Pt loading of 20 wt% (26 mV at 10 mA cm$^{-2}$, inset of Fig. 3a), and much higher than the commercial Pt/C (20 wt%) with same element Pt loading on the working electrode (Supplementary Fig. 16c, Pt loading is 1.31 μg cm$^{-2}$, different total loading). In contrast, Pt-NPs/CoNC shows a decreased HER activity compared to that of Pt-ACs/CoNC albeit with a higher Pt loading (1.47 wt% vs. 0.52 wt%), requiring a higher overpotential (42 mV) to achieve the same $j$ of 10 mA cm$^{-2}$. This phenomenon is ascribed to the aggregated Pt species in Pt-NPs/CoNC composite that has limited the accessible active sites, as well as the absence of interactions between Co and Pt atoms as revealed by the XPS and XAS results. The mass

activities of Pt-SAs/CoNC, Pt-ACs/CoNC, Pt-NPs/CoNC and the commercial Pt/C catalysts at an overpotential of 50 mV are calculated based on the linear sweep voltammograms (LSV) results in Fig. 3a and their corresponding Pt loadings. As shown in Fig. 3b, Pt-ACs/CoNC exhibits the highest mass activity of 28.6 A mg$^{-1}$Pt compared to those of Pt-SAs/CoNC (18.8 A mg$^{-1}$Pt) and Pt-NPs/CoNC (4.9 A mg$^{-1}$Pt). In addition, the mass activity for Pt-ACs/CoNC is 40 times higher than the Pt/C catalysts (0.7 A mg$^{-1}$Pt, Fig. 3b) with the same total loading and at least more than 6 times superior to the Pt/C catalysts with optimized mass activity (4.5 A mg$^{-1}$Pt, Supplementary Fig. 17), further demonstrating the Pt species in Pt-ACs/CoNC possess the highest intrinsic catalytic activity towards HER in acidic media[37]. With such a high activity and low Pt loading (0.52 wt%), Pt-ACs/CoNC is an active and cost-effective catalyst for hydrogen production in acidic media. Moreover, Pt-ACs/CoNC shows a low Tafel slope of 27.7 mV dec$^{-1}$, slightly lower than the commercial Pt/C catalyst (29.8 mV dec$^{-1}$) and Pt-NPs/CoNC (35.1 mV dec$^{-1}$), revealing its fastest HER kinetics (Fig. 3c). Pt-SAs/CoNC, on the other hand, exhibits a relatively higher Tafel slope of 64.5 mV dec$^{-1}$, suggesting slower kinetics for HER. The varied value of Tafel slopes suggests that HER on Pt ACs and Pt SAs may proceed through different

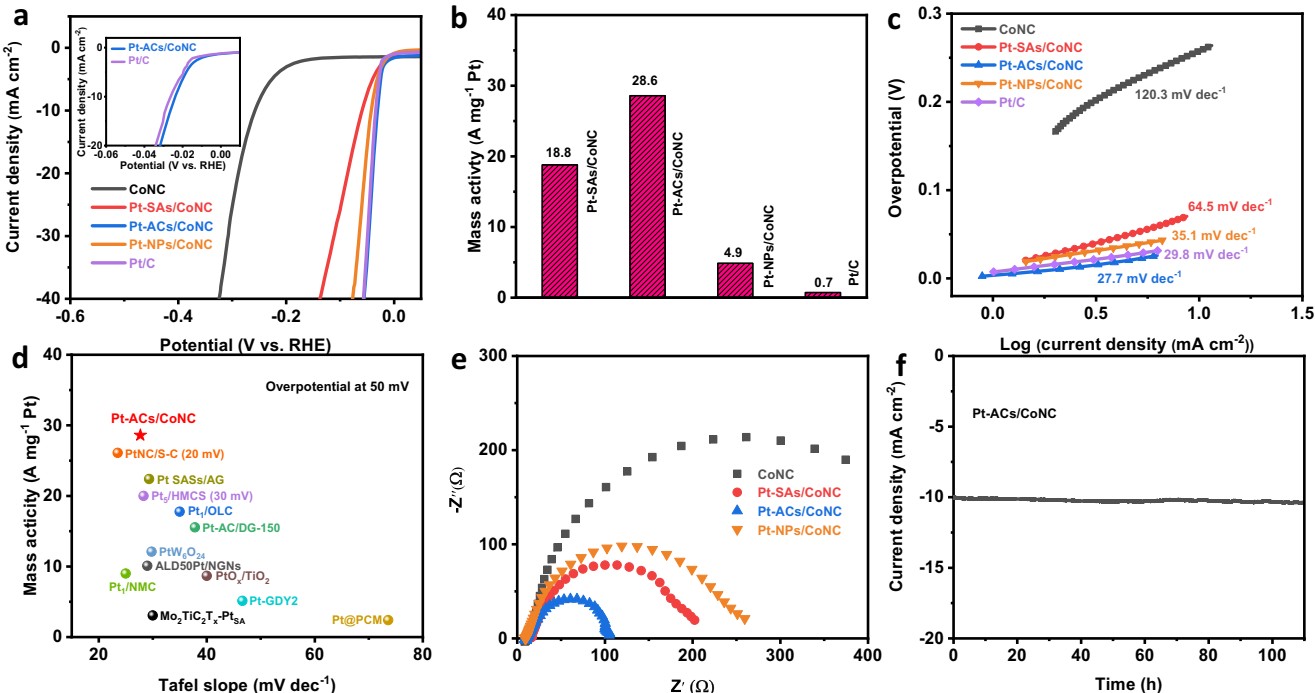

**Fig. 3 Electrocatalytic performance of Pt-ACs/CoNC and reference HER catalysts. a** HER polarization curves of CoNC, Pt-SAs/CoNC, Pt-ACs/CoNC, Pt-NPs/CoNC and Pt/C with same total loading on the working electrode with same geometric surface area in 0.5 M $H_2SO_4$ solution. **b** Mass activity of state-of-the-art Pt/C, Pt-SAs/CoNC, Pt-ACs/CoNC and Pt-NPs/CoNC at the overpotential of 50 mV with same total loading on the working electrode. **c** Corresponding Tafel slope. **d** Comparison of Tafel slope and mass activity for various Pt-based HER catalysts in 0.5 M $H_2SO_4$ solution. Values were plotted from references (Supplementary Table 2). **e** EIS Nyquist plots of CoNC, Pt-SAs/CoNC, Pt-ACs/CoNC and Pt-NPs/CoNC obtained at the overpotential of 20 mV. **f** Chronoamperometry curve of Pt-ACs/CoNC obtained at the overpotential of 27 mV.

reaction pathways[8,38]. The above merits of Pt-ACs/CoNC, including high mass activity and low Tafel slope, are superior to most reported Pt single atom or clusters-based HER catalysts (Fig. 4d and Supplementary Table 2).

The electrochemically active surface area (ECSA) was characterized by measuring the double-layer capacitance ($C_{dl}$) of the electrocatalysts within a non-Faradaic potential range instead of using hydrogen sorption peaks due to the low amount of Pt species (no obvious hydrogen sorption peaks can be detected as shown in Supplementary Fig. 18)[39]. It can be seen from Supplementary Fig. 19 that even though Pt-ACs/CoNC exhibits the highest HER activity, its $C_{dl}$ is smaller than that of Pt-SAs/CoNC and CoNC, indicating the large ECSA is not the main factor governing its high catalytic performance[40]. Fig. 3e shows the Nyquist plots of CoNC, Pt-SAs/CoNC, Pt-ACs/CoNC and Pt-NPs/CoNC. The semicircular diameters of electrochemical impedance spectra of Pt-SAs/CoNC, Pt-ACs/CoNC and Pt-NPs/CoNC are much smaller than that of CoNC, suggesting the incorporation of Pt species on CoNC can indeed enhance the HER kinetics. Moreover, Pt-ACs/CoNC exhibits the smallest semicircular diameter, indicating its lowest impedance (both contact and transfer) as well as the fastest reaction kinetics for HER.

The results of the long-term stability test of Pt-ACs/CoNC are presented in Fig. 3f and Supplementary Fig. 20, showing negligible loss of performance over 100 h of operation at the $j$ of 10 mA cm$^{-2}$, 50 h at $j$ of 40 mA cm$^{-2}$ and 5000 LSV repeating scans, which further demonstrates the exceptional electrochemical durability of the Pt-ACs/CoNC composite. The characterizations, including HADDF-STEM images, XPS spectra and Nyquist plots of Pt-ACs/CoNC after the stability test in Supplementary Fig. 21 confirm its physical robustness as well as the conductivity, and the structure of Pt ACs is well maintained after long-term

HER tests, demonstrating the high stability of Pt ACs supported by the atomically dispersed Co single atoms on carbon substrate[37]. Moreover, the ICP-MS test for the collected $H_2SO_4$ solution after the long-term test shows a negligible amount of Pt and Co (< 3.43 and 99.4 ppb, respectively) dissolved in the solution. All these results demonstrate the high stability of the Pt-ACs/CoNC composite in catalyzing HER in the acidic environment.

We further carried out a series of experiments to systematically investigate the actual active sites in Pt-ACs/CoNC for HER. The drastic enhancement of HER catalytic activity of Pt-ACs/CoNC over CoNC in Fig. 3a has already suggested that the incorporated Pt ACs might be responsible for hydrogen generation. To further confirm this, the HER performances of CoNC, Pt-SAs/CoNC and Pt-ACs/CoNC were evaluated in 0.5 M $H_2SO_4$ solution containing 5 mM of KSCN. It is reported that the SCN$^-$ ions tend to bind the metal species in catalysts, thereby lowering their activity in electrocatalysis[34,35,41]. As shown in Supplementary Fig. 22a, the introduction of SCN$^-$ has no effect on the HER performance of CoNC, indicating that the isolated Co atoms may not be the actual active sites. Also, as displayed in Supplementary Fig. 23, the bare nitrogen-doped carbon (NC, see the preparation details in Methods) is a fairly poor catalyst for HER in acidic media. Therefore, it is plausible to deduce that in CoNC, the non-metal atoms, such as N/C/O, adjacent to the Co atoms are electronically modified and activated and are acting as the active sites for HER. The HER performance of both Pt-ACs/CoNC and Pt-SAs/CoNC, on the other hand, decreases significantly in the presence of SCN$^-$ ions, with the overpotential at 10 mA cm$^{-2}$ increasing drastically from 24 to 169 mV, and 59 to 240 mV, respectively (Supplementary Fig. 22a, b). This reveals that the atomic Pt species are responsible for the high HER activity obtained with these two composites. Intriguingly, the poisoned Pt-ACs/CoNC still shows

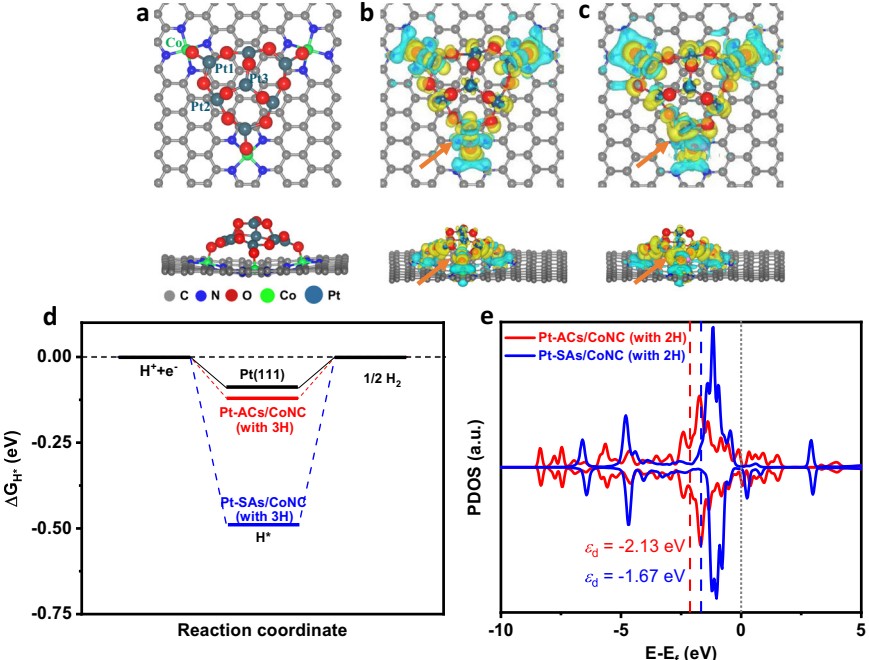

**Fig. 4 DFT calculation results. a** The top and side view of the atomic structure of Pt-ACs/CoNC. **b** the top and side view of deformation electronic density of Pt-ACs/CoNC. The yellow region represents charge accumulation and the blue region indicates charge depletion; the isosurface value is 0.001 e/Bohr[3]. **c** the top and side view of deformation electronic density of Pt-ACs/CoNC with O bridge occupied with H* (the orange arrow). **d** The $\Delta G_{H*}$ of HER at the equilibrium potential for Pt-ACs/CoNC, Pt-SAs/CoNC and commercial Pt/C. **e** The PDOS of d orbitals of active Pt atoms on Pt-ACs/CoNC and Pt-SAs/CoNC. The dotted grey line indicates the Fermi level, and the d-band centers ($\varepsilon_d$) are denoted by dashed lines.

much better HER performance than that of CoNC (overpotential 169 mV vs. 256 mV at 10 mA cm$^{-2}$), whereas the HER activity of Pt-SAs/CoNC is similar to that of CoNC (overpotential 240 mV vs. 256 mV at 10 mA cm$^{-2}$), suggesting that additional functional active sites are present in Pt-ACs/CoNC other than Pt and N/C atoms. We assign this to the linker O atoms in Pt-O-Pt unit, which may provide additional adsorption capability of H$^+$ during HER process. These adsorbed H$^+$ can be transferred to adjacent Pt atoms, which in turn increases the overall HER performance[42].

Based on the above results, it is clear that the Pt-O-Pt units in Pt-ACs/CoNC are the main active sites for HER, where both Pt and O atoms are capable of adsorbing H$^+$ from the electrolyte[41]. The low valence state of Pt atoms in Pt-ACs/CoNC implies that less electrons have been transferred to the nearby O atoms, leading to a lower electron density on these O linkers. It significantly reduces the capability of O atoms to form stable onium cations with H$^+$. Therefore, the adsorbed H$^+$ can be easily desorbed from the O atoms and transferred to the surrounding Pt atoms. Furthermore, the higher electron density of Pt atoms in Pt-O-Pt units (XPS and XAS results) make them highly favourable for both H$^+$ adsorption and HER process[14]. As a result, the continuous adsorption of H$^+$ on Pt/O atoms and H transfer from O to adjacent Pt atoms dramatically accelerate the rate of H combination on Pt sites, allowing hydrogen evolution to proceed via a Tafel reaction process (Tafel slope of 27.7 mV dec$^{-1}$, DFT calculations also confirm the Volmer-Tafel catalytic pathway, Supplementary Fig. 24)[38]. In contrast, the isolated Pt site from Pt-SAs/CoNC adsorbs one H$^+$ from the electrolyte. Without the facilitation of O linkers to provide additional H$^+$ adsorption, the second H$^+$ tends to attach to the active H* atom on Pt sites to produce H$_2$ molecule, following a relatively slower Heyrovsky reaction pathway (Tafel slope of 64.5 mV dec$^{-1}$)[43]. Pt-NPs/CoNC possesses Pt NPs resembling commercial Pt/C catalyst, producing H$_2$ via a Tafel reaction (Tafel slope of 35.1 mV dec$^{-1}$). However,

the aggregation of these Pt NPs significantly reduces the Pt utilization, especially under the condition of low Pt content (around 1.5 wt%), which significantly reduces the overall activity of the catalyst. The above results evidently reveal that Pt-ACs/CoNC with Pt-O-Pt units can dramatically boost the electrochemical HER process.

**DFT calculations.** DFT calculations were performed to further elucidate the HER catalytic mechanism of Pt-ACs/CoNC. A Pt$_7$O$_{12}$ cluster was constructed on a CoN$_4$-carbon surface to simulate the atomic structure of Pt-ACs/CoNC (Fig. 4a, the other constructed atomic structures are shown in Supplementary Fig. 25)[18]. After structural optimization (refer to the Supporting Information for details), each Pt atom is stabilized by three O atoms, and three of these Pt atoms interact with Co SAs through O atom at the interface. This is consistent with the fitting results of EXAFS in Fig. 2e. Based on the proposed structure, we consider two potential sites as active HER catalytic centers in the constructed Pt-ACs/CoNC system, namely the Co-O-Pt interface and the Pt-O-Pt unit. The charge density distributions in Fig. 4b show that significant electron transfer occurred at the Co-O-Pt interface, resulting in reduced electron density around these Co/Pt atoms and strong interaction between Pt ACs and CoNC substrate. Although this helps to stabilize Pt ACs on CoNC, the reduced electron density of metal species is not beneficial to the HER activity (the free energies of H* ($\Delta G_{H*}$) on these Co and Pt atoms are shown in Supplementary Fig. 26)[14]. The constructed atomic structure of Pt-SAs/CoNC is shown in Supplementary Fig. 27 with Pt-O-Co bond, which is similar to the Co-O-Pt interface in Pt-ACs/CoNC. The strong interaction within Pt-O-Co bond induces the relatively lower HER activity of Pt-SAs/CoNC compared with Pt-ACs/CoNC. In contrast, the charge density distributions of Pt-O-Pt species in Pt-ACs/CoNC reveal less charge delocalized from Pt to the nearby O atoms, making

these Pt atoms with higher electron density (consistent with the XPS and XANES results in Fig. 2 and Supplementary Fig. 11c). This strongly supports our proposed catalytic mechanism, making both Pt and O atoms synergistically beneficial to hydrogen generation[42]. Moreover, when O site in Pt-O-Pt unit is occupied with H*, the electron density around the nearby Pt will be further enhanced, resulting in optimized H* adsorption energy (Fig. 4c). Therefore, the kinetics of HER can be significantly boosted on these Pt atoms with additional $H^+$ adsorption on surrounding O linkers. The calculated $\Delta G_{H*}$ for Pt-ACs/CoNC decreases from $-0.64$ eV to $-0.12$ eV as the H* coverage increases from 1 to 3 (Supplementary Fig. 28 and Fig. 30), which is close to the ideal HER catalyst (Fig. 4d, $\Delta G_{H*}$ of Pt (111) is $-0.09$ eV). In contrast, the calculated $\Delta G_{H*}$ of Pt-SAs/CoNC (Supplementary Fig. 29 and Fig. 30) only decreases slightly from $-0.58$ eV to $-0.49$ eV with H* coverage varies from 1 to 3, echoing its inferior catalytic activity in HER as compared to Pt-ACs/CoNC. This suggests that Pt-O-Pt units in Pt-ACs/CoNC with high H* coverages play an important role in the HER process. Furthermore, Pt-O-Pt units could also facilitate the desorption of $H_2$ molecules with a free energy of nearly 0 (0.078 eV) based on the DFT calculation. This indicates that $H_2$ can be easily desorbed from the active sites in Pt-ACs/CoNC, leading to the further enhancement of the HER performance.

In order to further elucidate why Pt-O-Pt cluster exhibits superior activity, the detailed electronic structures are explored. The calculated projected density of states (PDOS) of d orbitals and the d-band center ($\varepsilon_d$) of active Pt atoms on Pt-ACs/CoNC and Pt-SAs/CoNC are shown in Fig. 4e. The results show a significant left shift of PDOS for the d orbitals of Pt atom on Pt-ACs/CoNC compared to that on Pt-SAs/CoNC. The d-band center of Pt atom on Pt-ACs/CoNC is further away from the Fermi level than that on Pt-SAs/CoNC by 0.46 eV ($\varepsilon_d$ are $-2.13$ and $-1.67$ eV for Pt-ACs/CoNC and Pt-SAs/CoNC, respectively), resulting in weaker binding to H*, which would enhance the HER activity of Pt-ACs/CoNC. Therefore, the Pt-O-Pt units in Pt-ACs/CoNC indeed act as the active sites that markedly facilitate the HER catalytic process.

## Discussion

Apart from Pt ACs, the use of CoNC substrate can also facilitate the formation of other noble metal-based atomic clusters due to the presence of numerous Co SAs as the anchoring sites, as well as the construction of strong interactions between Co SAs and noble metal ACs that can restrict their unwanted aggregations. Ru-ACs/CoNC and Ir-ACs/CoNC have been prepared by the same synthetic method as Pt-ACs/CoNC (details are presented in the experimental section), and the successful achievement of metal ACs has been confirmed by a suite of physical characterizations (Fig. 5 and Supplementary Figs. 31, 32). As shown in Fig. 5a, c, the tiny clusters with an average size of less than 1 nm and single atoms with less contrast in HAADF-STEM images indicate that Ru and Ir ACs are stabilized on the CoNC substrate. Elemental mapping results in Fig. 5b, d show the existence of Ru, Co, N, C and Ir, Co, N, C with homogeneous distribution in Ru-ACs/CoNC and Ir-ACs/CoNC, respectively. The formation of interactions between Co SAs and Ru/Ir ACs has been revealed by the XPS results (Fig. 5e), reflected by the shifted Co 2p peaks as compared to that of CoNC. All the above results demonstrate that Ru-ACs/CoNC and Ir-ACs/CoNC share a similar structure to that of Pt-ACs/CoNC. Both Ru-ACs/CoNC and Ir-ACs/CoNC exhibited an obviously enhanced HER catalytic activity compared to the pure CoNC in acidic media by reducing the overpotential from 256 mV to 72 mV and 50 mV for the j of 10 mA cm$^{-2}$, respectively (Fig. 5f). Moreover, the results show Ru-ACs/CoNC,

Pt-ACs/CoNC and Ir-ACs/CoNC also exhibit much superior HER capability compared to CoNC in 1 M KOH (Fig. 5g). In particular, Ru-ACs/CoNC display the highest HER performance with a relatively low overpotential of 40 mV to achieve a j of 10 mA cm$^{-2}$, which is even better than the commercial Pt/C (46 mV for 10 mA cm$^{-2}$). The HER performance further illustrates that the incorporation of noble metal-based ACs on CoNC can accelerate the HER process in both acidic and alkaline solutions.

In conclusion, we report atomic-layered Pt ACs containing Pt-O-Pt units uniformly dispersed on Co single atoms and N co-doped porous carbon materials, which exhibit excellent HER catalytic performance. The isolated $CoN_4$ species provide sufficient anchoring sites to stabilize the Pt clusters. The strong interaction between $CoN_4$ and Pt ACs leads to charge redistribution, accounting for their superior stability (over 100 h of HER). XPS and XAS results demonstrate that the obtained Pt ACs are in a low oxidation state, which is beneficial for hydrogen generation. Particularly, the mass activity of Pt-ACs/CoNC is over 40 times higher than that of a commercial Pt/C catalyst, demonstrating the intrinsic catalytic capability of Pt-ACs/CoNC. The Pt ACs consisting Pt atoms bridging with O atoms possess intriguing electronic properties, and the $\Delta G_{H*}$ of Pt-O-Pt unit decreases significantly with the enhancement of H coverage, highly promoting the hydrogen/proton adsorption and lowering the kinetic energy barrier via minimal Pt incorporation. Our studies highlight the significance of creating Pt-O-Pt atomic clusters supported on atomically dispersed Co atoms for boosting the HER process. These findings also shed light on the designed synthesis of high-performance catalysts with delicately controlled active centers. Moreover, the constructed unique catalytic system may serve as a powerful platform for future studies of many other energy conversion reactions and organic synthesis, such as oxygen reduction reaction, methanol/alcohol oxidation reaction, n-butane dehydrogenation reaction, to name a few.

## Methods
### Materials preparation
*Synthesis of CoNC.* CoNC was fabricated by an optimized polymerization-pyrolysis method followed by a chemical leaching process. Typically, 3 g of o-phenylene-diamine (o-PD) monomer and 15.4 ml of colloid silica solution (LUDOX HS-30) was mixed in 20 ml of deionized water. Then 2 ml of hydrochloric acid (HCl, 1 M) was added into the above solution slowly under continuous stirring to assist the well-dispersion of o-PD. Afterward, the solution was cooled down to 5 ºC by a cycling water system. After the solution was stabilized at 5 ºC, 1.58 g of cobalt nitrate (Co(NO$_3$)$_2$) in 5 ml deionized water was added to the above solution, followed by introducing ammonium persulfate (NH$_4$)$_2$S$_2$O$_8$) solution (0.96 g in 10 ml deionized water) to trigger the polymerization process of o-PD. The reaction was allowed to run for 24 h, and then the samples were collected by rotation evaporation. The achieved samples were calcined at 900 ºC for 3 h under Ar atmosphere. The final product, CoNC, was obtained by removing the silica template and possible metal nanoparticles by 2 M NaOH and 0.5 M H$_2$SO$_4$ hot solutions, respectively, and another annealing at the same conditions at 900 ºC.

*Synthesis of Pt species on CoNC (Pt/CoNC).* Pt-ACs/CoNC was prepared by a conventional wet-impregnation method. In the typical synthesis process, 50 mg CoNC was dispersed in 10 ml ethanol by ultrasonication for 30 min. Afterward, 504 μl of platinum(II) bis(acetylacetonate) (1 mg ml$^{-1}$, ethanol) was added to the above solution under vigorous stirring for another 24 h. The resulting mixture was dried by evaporating the ethanol under 80 ºC. Then, the final sample was achieved by calcinating the dried samples at 300 ºC for 2 h under Ar atmosphere. Pt-SAs/CoNC and Pt-NPs/CoNC were prepared by a similar method with the alternation of the amount of platinum(II) bis(acetylacetonate) solutions to 101 and 1008 μl, respectively.

*Synthesis of Ru-ACs/CoNC and Ir-ACs/CoNC.* Ru-ACs/CoNC and Ir-ACs/CoNC were prepared by a similar method to Pt-ACs/CoNC. In the typical synthesis process, 50 mg CoNC was dispersed in 10 ml ethanol by ultrasonication for 30 min. Afterward, 960 μl of ruthenium(III) acetylacetonate and 636 μl of iridium acetylacetonate (1 mg ml$^{-1}$, ethanol) were added to the above solution under vigorous stirring for another 24 h. The resulting mixture was dried by evaporating the ethanol under 80 ºC. Then, the final samples, Ru-ACs/CoNC and Ir-ACs/CoNC,

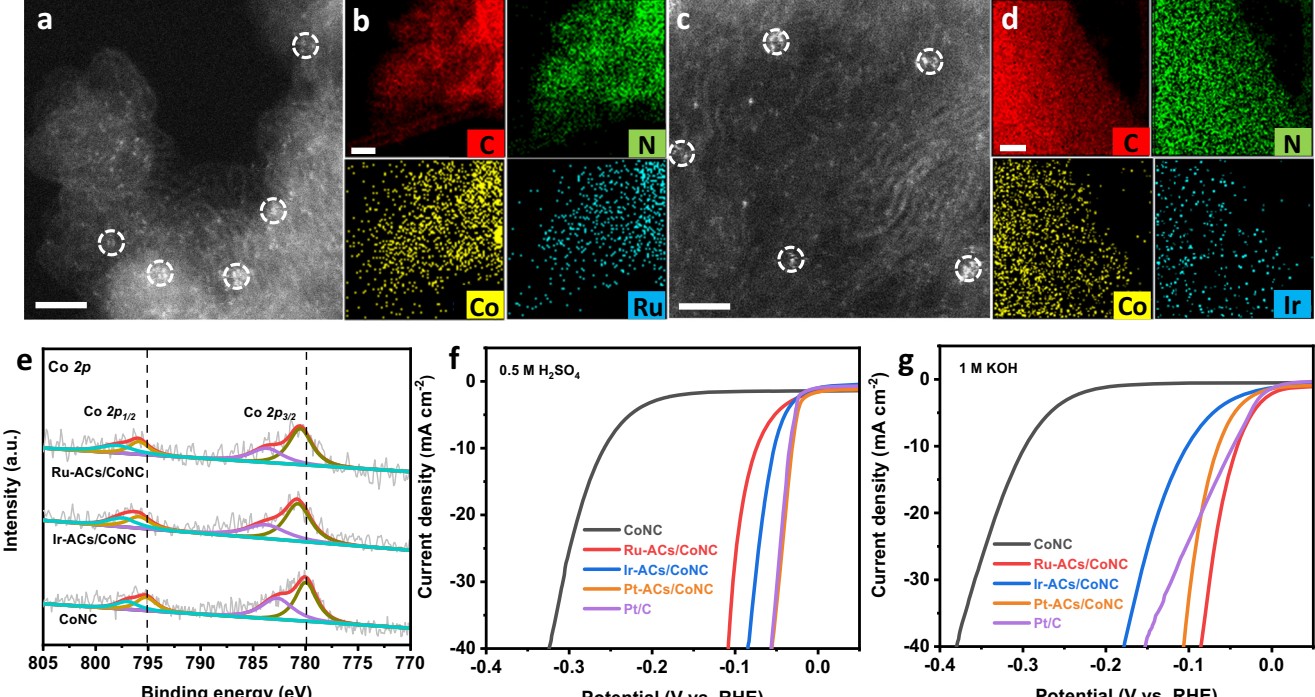

**Fig. 5 Characterization and HER performance of Ru-ACs/CoNC and Ir-ACs/CoNC. a, b** STEM image and elemental mapping of Ru-ACs/CoNC. Scale bar: 2 and 50 nm. **c, d** STEM image and elemental mapping of Ir-ACs/CoNC. Scale bar: 2 and 50 nm. **e** The high-resolution XPS spectra of Co 2p in CoNC, Ru-ACs/CoNC and Ir-ACs/CoNC. **f, g** HER polarization curves of CoNC, Ru-ACs/CoNC, Ir-ACs/CoNC, Pt-ACs/CoNC and Pt/C in 0.5 M H$_2$SO$_4$ and 1 M KOH, respectively.

were achieved by calcinating the dried samples at 300 °C for 2 h under Ar atmosphere.

**Characterization.** XRD spectra were recorded on Bruker D8 Discover XRD with the 2 $\theta$ range from 10 to 80° and a scanning step of 0.04° per second. The morphological characterization and EDS analysis of the as-prepared samples were investigated by TEM, Model JEOL JEM-F200, STEM, JEOL JEM-ARM200F and FEI Themis-Z Double-corrected 60–300 kV. XPS spectra were acquired on an ESCALAB250Xi (Thermo Scientific, UK). Raman spectra were recorded on an inVia Renishaw Raman spectrometer system (HR Micro Raman spectrometer, Horiba JOBIN YVON US/ HR800 UV) with a wavelength laser of 632.8 nm. ICP-MS was performed on PerkinElmer Nexion. The BET surface area of the prepared samples was measured by using experimental points at a relative pressure of P/P$_0$ = 0.05–0.25. The XANES and EXAFS at Pt L-edge and Co K-edge were determined in a transition mode at the Synchrotron Radiation Research Center, Singapore. Pt foil, Co foil, PtO$_2$, CoO and Co$_3$O$_4$, were used as reference samples. ATHENA software was used to analyse the obtained data based on the standard procedures. The fitting results were acquired in the $k^3$-weighted EXAFS oscillation using the module ARTEMIS of IFEFFIT at the range of 0–6 Å.

**Electrochemical Measurements.** HER performance tests in 0.5 M H$_2$SO$_4$ and 1 M KOH solution were conducted in a three-electrode configuration by using the electrochemical workstation of CHI 660E, CH Instrument. Rotating disk electrode (RDE), graphite rod and saturated calomel electrode (SCE) were used as working, counter and reference electrodes, respectively. The reference SCE was calibrated vs. RHE, $E_{RHE} = E_{SCE} + 0.059 \times pH + 0.241$ V. The electrode slurry was fabricated by mixing 4 mg of the prepared catalysts, 80 $\mu$l of Nafion (5 wt %) and 1 ml solvent (1:1 v/v water/ethanol) and then sonicated for 30 min to form a dispersion. 5 $\mu$l of the dispersion was dropped onto the RDE, followed by drying at room temperature. The total loading of Pt-ACs/CoNC was 262 $\mu$g cm$^{-1}$ and the Pt loading was 1.31 $\mu$g cm$^{-2}$. All other electrodes were prepared by depositing the same total mass loading of catalysts (262 $\mu$g cm$^{-1}$) on the RDE with the same geometric surface area using an identical method (Supplementary Table 2). The Pt loading of commercial Pt/C (20 wt%) was 52.4 $\mu$g cm$^{-2}$. The commercial Pt/C with different Pt loading on the working electrodes was prepared by diluting the above prepared ink and 5 $\mu$l of the diluted dispersion was dropped onto the RDE. The polarization curves were obtained with a scan rate of 5 mV s$^{-1}$, which was all corrected for the $i$R contribution within the cell. Cyclic voltammetry (CV) measurements were performed for Pt-SAs/CoNC and Pt-ACs/CoNC in the potential window of 0-0.5 V (vs. RHE). CV was conducted to measure the electrochemical capacitance with the scan rates from 10 to 100 mV s$^{-1}$ and potential centered at open circuit potential.

The electrochemical impedance spectra (EIS) spectra were acquired with an amplitude of 5 mV, a frequency from 10$^6$ to 0.01 Hz, and an overpotential of 20 mV. The stability of Pt-ACs/CoNC was tested by repeating LSV running for 5000 cycles (accelerated scan rate 100 mV s$^{-1}$) with the potential range between 0.10 and −0.15 V vs. RHE and the current-time plots were achieved at overpotentials of 27 mV and 63 mV to achieve the current density of 10 and 40 mA cm$^{-2}$. After the stability test, the samples were washed, collected and stored in the vacuumed desiccator to avoid the reoxidation of Pt species.

Tafel slopes were obtained by linear fitting the plot derived from the logarithm of current density vs. overpotential. The Tafel slopes were determined from Tafel equation:

$$\eta = b \log j + c \tag{1}$$

where b is the Tafel slope, $\eta$ is the overpotential, j is the current density, c is the intercept.

The mass activity of the catalysts was calculated according to the following equations:

$$\text{mass activity} = j/m_{Pt} \tag{2}$$

where j is the measured current density (mA cm$^{-2}$), m is the catalyst loading based on element Pt (mg cm$^{-2}$).

**DFT Calculations.** All of the spin-polarized DFT calculations were performed using the VASP program[44,45], which used a plane-wave basis set and a projector augmented wave method (PAW) for the treatment of core electrons[46]. The Perdew, Burke, and Ernzerhof exchange-correlation functional within a generalized gradient approximation (GGA-PBE)[47] was used in our calculations, and the van der Waals (vdW) correction proposed by Grimme (DFT-D3)[48] was employed due to its good description of long-range vdW interactions. For the expansion of wavefunctions over the plane-wave basis set, a converged cutoff was set to 500 eV. Self-consistent-field (SCF) calculations were performed with an electronic structure iteration of 1 × 10$^{-5}$ eV on the total energy, and the atomic positions were optimized until the forces were below 0.01 eV/Å during structural optimization.

Ab initio molecular dynamics (AIMD) simulations were also carried out using VASP. The wavefunctions were expanded with a kinetic energy cut-off value of 450 eV. Otherwise, similar parameters mentioned above were employed. MD simulations were carried out using the NVT ensemble using a Nose-Hoover thermostat, and the temperature was ramped up from 10 K to 310 K over a time period of 4 ps. A time-step of 1 fs was used in all our simulations.

In order to simulate CoN$_4$ moieties embedded in a basal carbon plane (CoNC), a 10 × 6 carbon supercell with periodic boundary conditions was used, and then,

six carbon atoms were removed to create a $CoN_4$ moiety. Here, we created three $CoN_4$ moieties embedded in a basal carbon plane. A $Pt_7O_{12}$ and a $Pt_1O_3$ were constructed on $CoN_4$-carbon surface to simulate Pt ACs and the Pt SAs, respectively, as per experimental findings. We have also generated additional atomic structures, i.e., $Pt_{11}O_{16}$ and $Pt_{18}O_{24}$, on a single $CoN_4$ moiety embedded in a basal carbon plane to explore alternative plausible AC structures. The stability of these ACs was tested using AIMD simulations. The vacuum space was set to larger than 25 Å in the z direction to avoid interactions between periodic images. The Brillouin zone integration was performed on the $(2 \times 2 \times 1)$ Monkhorst-Pack k-point mesh[49].

The overall HER mechanism was evaluated with a three-state diagram consisting of an initial $H^+$ state, an intermediate $H^*$ state, and $1/2$ $H_2$ as the final product. The free energy of $H^*$ ($\Delta G_{H^*}$) was proven to be a key descriptor to characterize the HER activity of the electrocatalyst. An electrocatalyst with a positive value leads to low kinetics of adsorption of hydrogen, while a catalyst with a negative value leads to low kinetics of release of hydrogen molecule[50]. The optimum value of $|\Delta G_{H^*}|$ should be zero; for instance, this value for the well-known highly efficient Pt catalyst is near-zero as $|\Delta G_{H^*}| \approx 0.09$ eV[7]. The $\Delta G_{H^*}$ is calculated as[49]

$$\Delta G_{H^*} = \Delta E_{H^*} + \Delta E_{ZPE} - T\Delta S_H \quad (3)$$

where $\Delta E_{H^*}$ is the binding energy of adsorbed hydrogen, and $\Delta E_{ZPE}$ and $\Delta S_H$ are the difference in ZPE and entropy between the adsorbed hydrogen and hydrogen in the gas phase, respectively. As the contribution from the vibrational entropy of hydrogen in the adsorbed state is negligibly small, the entropy of hydrogen adsorption is $\Delta S_H \approx -\frac{1}{2}S_{H_2}$, where $S_{H_2}$ is the entropy of $H_2$ in the gas phase at the standard conditions. In this work, we defined the $\Delta G_{H^*}$ values as $\Delta E_H + 0.24$eV for all catalysts[51].

## Data availability

The data supporting this study are available within the paper and the Supplementary Information. All other relevant source data are available from the corresponding authors upon reasonable request. Source data are provided with this paper.

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

## Acknowledgements

We thank UNSW Mark Wainwright Analytical Center for providing access to the SEM, TEM, XPS and other facilities. The authors acknowledge use of JEOL JEM-ARM200F STEM within the University of Wollongong (UoW) Electron Microscopy Centre and XAS measurements at the Synchrotron Radiation Research Center, Singapore. Dr. Jinqiang Zhang and Dr. Pengfei Ou from the University of Toronto (UofT) are acknowledged for their valuable discussions. This work was supported by the Australian Renewable Energy Agency (ARENA 2018/RND014) and Australian Research Council (ARC ITRP Scheme-IC200100023). X. L. and P.V.K. thank the UNSW Scientia Scheme for financial supports. P.V.K acknowledges the ARC DECRA Fellowship (DE210101259). This research was also undertaken with the assistance of resources provided by the National Computing Infrastructure (NCI) facility at the Australian National University; allocated through both the National Computational Merit Allocation Scheme supported by the Australian Government and the Australian Research Council grant LE190100021 (sustaining and strengthening merit-based access at NCI, 2019–2021).

## Author contributions

R.A., X.L., and Z.H. directed the project. X.L., Z.H., and Y.Z. designed the experiments. Y.Z., X.Z., T.W., and D.C. synthesized the catalysts. Y.Z. carried out characterizations and performed the electrochemical experiments. S.X., H.Y., and J.J. performed the XAS experiments and analysis the data. X.T., P.V.K., S.S., and X.X.L. performed computational modeling. J.P., W.J., and Z.M. analysed the data. X.L., Z.H., R.A., Y.Z., and X.T. co-wrote the manuscript.

## Competing interests

The authors declare no competing interests.
