## [Peer Review File · Nature Communications]

Modulating Pt-O-Pt atomic clusters with isolated cobalt atoms for enhanced hydrogen evolution catalysisREVIEWER COMMENTS

Reviewer #1 (Remarks to the Author):

The manuscript by Zhao et al. reported a design and synthesis of a class of M-O-M/CoNC catalysts that are highly active for HER reaction under acidic conditions. The highlights of the work include the Co-induced nucleation of Pt-O-Pt atomic clusters that are believed to be the effective catalytic center for the HER reaction. A systematic comparison was also made between the Pt-O-Pt atomic cluster, the Pt single-atom and the Pt nanoparticle, and the fundamental research work reveals that the Pt-O-Pt type of structure is the more suitable catalytic moiety for the targeted reaction. While the work is potentially suitable for publication by Nature Communications, I feel the rigor needs being improved in several key aspects of the paper.

1) The computationally proposed Pt-O-Pt atomic structure is very particular in terms of its own structure and how the Co anchoring sites are located relative to the platinum species. Such a kind of precision and uniformity does not exist according to the HAADF-STEM images. Perhaps multiple plausible computational models should be proposed and examined to show the superiority of the Pt-O-Pt atomic cluster over the single-atom and nanoparticle counterparts are generally applicable.

2) I am a bit concerned if the present CoNC platform is suitable for the general interrogation of single-atom Pt vs. Pt-O-Pt catalytic center? To answer such a type of fundamental question, usually researchers would choose the uniform or unbiased support/ substrate system to carry the supported catalytic centers to conduct the investigation. However, in this work, the extent of interaction between the platinum atoms and the cobalt atoms are bound to be different as the function of platinum structures/ sizes, so the varied catalytic performance is no longer a sole function of interatomic Pt-O-Pt interactions. The role of Co should be further understood in this work. Notably, from Ref. 15, the single-atom Pt catalyst seems even more active than the best catalyst reported in this work. This further manifest that the cobalt is having a role to play. The least to say is the role of Co in modifying Pt activity should be further understood.

3) The synthesis, if without further justification for the Pt precursor impregnation, is somewhat lacking elegance to truly form uniform structures with atomic precision. For example, how do we make sure the Co terminal can attract all the Pt forming exclusive types of moieties as single atom, atomic cluster, or nanoparticles? What is the surface bonding environment that promotes the exclusive formation of either one of the three representative species?

4) If the formation of increasingly aggregated platinum species is a sole function of higher platinum loading, why the Pt NPs/CoNC sample lacks the attributes of the Pt ACs/CoNC sample according to HER activity data and the Tafel slope? Even if the Pt NPs are having zero activity for the given testing conditions, one would expect the remainder of the Pt ACs in the same sample to perform equally well as in the Pt ACs/CoNC sample.

5) Some explanations are required for the noticeable shifts of Co peaks in the XPS data reported in Fig. 2. The authors also claimed that supplementary Figure 3 reveal that isolated Co atoms are negatively charge. How? Also, the fitting of XPS data of core level Co-2p needs reconsideration. "The higher electron density of these O atoms makes them easier to capture the incorporated Pt ions" line-111-114. The above sentence needs further explanation and proof.

6) Line-303-305: "promising HER catalysts for next generation PEM electrolyzers". Since the study was done on GCE (< 3mm dia), this statement should be reserved until and unless authors have done- (1) large area study with catalysts deposited on gas diffusion electrodes. (2) The Methanol and CO tolerance study.

7) Authors should mention in the caption the details of chronoamperometry as well as the accelerated LSV cycle study. At what voltage it was performed. What is the potential window for LSV cycling? At what voltage the EIS study was done?

8) Considering the fact that HER is a reductive reaction and the platinum species, especially when being small, are very sensitive to air reoxidation. Many of the key characterization pertaining to the oxidation state of Pt-O-Pt should be carried out systematically in situ or with good sealing after reaction.

Reviewer #2 (Remarks to the Author):

In this manuscript, Zhao et al. use a combination of. Various experimental methods and computation to evaluate Pt-O-Pt clusters bound to CoN4C sites on graphitic carbon that they have synthesized as well as Pt nanoparticles on CoN4Co and Pt single atom catalysts supported on CoN4Co and Pt supported on C supports. These results represent an important advance and will be of significance to the electrocatalysis field. Although I was impressed with the advances made in this ms, it also suffers from a number of flaws that must be addressed before it is publishable in a journal of the quality of Nat. Comm. There are several technical questions that should be addressed, and the writing needs considerable improvement.

Strengths:

1. The improved HER specific activity the authors were able to achieve with the "atomic cluster" catalysts compared to both SA and NP catalysts (Fig 3b). There is clearly something unique about these active sites beyond just being monodisperse.

Suggestion: Some sort of error analysis (like error bars showing standard deviations) would make this aspect even stronger.

2. The authors performed an extensive amount of characterization of active sites of the AC catalyst. This provides insight into why these ACs are more active than the SA or NP catalysts. The XANES results showing the lower Pt valence state for the atomic cluster catalysts are particularly interesting.

Suggestion: The authors claim that Pt bound to O in the Pt-O-Pt configurations has a lower oxidation state than SA Pt catalysts. The reasoning for this is not clear as O bound to Pt usually raises the oxidation state of Pt.

3. The overall advance and narrative describing the advance is simple and relatively clear. They authors synthesize dispersed Pt catalysts that are more active than other monodisperse catalysts due to specific interactions between the Pt atoms and CoNC support. These interactions are well-explored with experimental approaches and some supporting computational analysis is provided. The end result is a more efficient HER catalyst.

Suggestion: Although the story is relatively clear, the manuscript suffers from numerous grammatical errors and awkward and confusing phrasing. This makes the paper unnecessarily difficult to read and lowers its overall quality.

Suggestion: Although the story is simple, to increase the impact of this work and make it appropriate for a Nature journal, it should provide some perspective into how this approach can have broader impact beyond HER by supported Pt catalysts.

Weaknesses:

4. Figs 3a and 3c show the AC and generic Pt/C catalyst performed almost identically. There is no indication of experimental error, so it's hard to believe that these are significantly different. Is this due to them reaching some sort of "maximum rate" for their experimental system (e.g., being transport limited?) or is it just a coincidence? Are there conditions/catalyst loadings where they can show a significant separation between performance of these two catalysts? Furthermore, the specific activity will be affected if they are reaching a systematic maximum rate.

Suggestion: The claim that the Pt-AC catalyst performs better needs to be supported by some sort of

error analysis.

5. The analysis of catalyst stability is limited to just the Pt-AC catalyst under one set of conditions where the catalyst is stable.

Suggestion: It would be useful to know the conditions under which these active site structures become unstable, so as to direct future use of these materials.

6. As mentioned above, the quality of the writing needs improvement. The grammar and phrasing is very poor throughout the ms (although less so in the computational section). This will greatly improve readability and avoid confusion.

Suggestions (beyond a careful edit and proofread by someone fluent in English):

Use simpler terms for the different catalysts. E.g., Pt-ACs instead of Pt-ACs/CoNC or Pt-SA instead of for Pt-SAs/CoNC. The support is the same throughout, so no need to repeat it every time, especially because it makes the text cumbersome to read.

The Pt-SAs is not defined in the ms and no figure of the atomic structure is included for it, so the reader can only guess exactly what this structure is.

The names of the clusters are not very descriptive. SA, AC and NP essentially only indicate size, but not coordination. This is especially an issue because the SA structure is not defined. AC is particularly non-descript because it doesn't indicate that this isn't just a small cluster of Pt atoms. Maybe a name like Pt-O-C or Pt-O-AC is better.

Technical Issues:

A very specific Pt-O-AC structure was used for DFT modeling – do other similar models give similar or different spectra, H* binding energies, etc.

The DFT model is very simplified relative to the actual system. It does not include the effect of bias (e.g., using a grand canonical description that allows the electron number to change to match the Fermi level to the applied potential) nor solvent. There is no justification for why a vacuum model should accurately describe what is happening at the electrified interface.

No barriers are calculated so suggestions about the kinetics based on the DFT computed-thermodynamics (e.g., H* binding free energies) is speculative.

At what reducing potential are Pt-O-Pt structures reduced to Pt (or other degradation process)? How close is the operating bias to potentials at which degradation occurs?

In the computational studies there is no indication of the Co center's interaction with something at the 6-position (e.g., from the first sublayer of the support). Justify this assumption.

In Fig 1 j distributions are giving for C, N, Co and Pt, but not for O, although it is provided in the SI for the NP and SA catalysts. Also, no statistics are presented for correlations in the distributions (e.g., between Co and Pt and between O and Pt).

Are other CoNCs present besides the CoN4C pyridinic site ? E.g., CoN3C or a pyrolic sites?

Are all N doped sites occupied by Co or might some N sites interact directly with Pt in the absence of Co (e.g., as in Muhich et al. 2013)?

There is no explanation given for very similar rates in Figs 3a and c between Pt-O-ACs and traditional Pt on C catalysts.

On line 310 the authors suggest that the different Tafel slopes suggest different HER pathways.

However, the different kinetics could be explained by different activation barriers on the different active sites. DFT computed barriers could resolve this.

Why are no results reported for more reducing potentials than 256 mV? In Fig 3c the current density was still increasing with larger overpotential.

Structures in Figs 4 b and c should not be rotated relative to 4a to make it easier to understand the density isosurface relative to the atomic structure.

Reviewer #3 (Remarks to the Author):

The manuscript from Zhao et al. presents the preparation route, physicochemical characterization and the performance toward hydrogen evolution reaction (HER) of a class of promising ultra-low loading Pt catalyst. The excellent performance and stability toward HER of the presented material composed of ultrasmall Pt atomic clusters compared to single atom or nanoparticulate catalysts is rationalized by the stabilization of the Pt clusters by CoN₄ species from the carbon support, and synergetic electronic effects from Pt cluster composition and metal-support interactions.

To support their hypothesis, the authors used an impressive combination of advanced characterization techniques completed with DFT calculations. In overall the manuscript is clear and convincing, I would thus recommend publication in Nature Communications after the authors consider some minor revisions.

1/In the microscopy part of the manuscript, the X-EDS maps with linescan analyses are not all convincing. From Figures 1 and S2, the fact that N, Co or Pt elements appear also out of the carbon matrix suggests the signal/noise ratio (counting time) may not be sufficient. Whereas it can be enough to support the overall homogenous repartition of such elements at the ~100 nm scale, the linescans Figure 1.i and S5 on one single atomic cluster <1nm needs precautions.

Can the authors show the X-EDS spectra restricted to the region of atomic cluster investigated and associated elemental map? If there is not enough resolution and/or intensity, there is no point in doing linescans.

2/ How do the authors extract sample height with single atom resolution from HAADF-STEM images in Figure 1.h?

3/Please add some legend for the color code used in cartoons in Figures S14 , 1.f, 4.

4/When Figure S3 is mentioned in the text, the reader has no idea what is the model used behind the fit of the EXAFS curve. Maybe Figure S14 should be merged with Figure S3.

5/ Please replace 'spectra' per 'patterns' in figure S28 caption.

Detailed Reviewer Response

Reviewer #1:

The manuscript by Zhao et al. reported a design and synthesis of a class of M-O-M/CoNC catalysts that are highly active for HER reaction under acidic conditions. The highlights of the work include the Co-induced nucleation of Pt-O-Pt atomic clusters that are believed to be the effective catalytic center for the HER reaction. A systematic comparison was also made between the Pt-O-Pt atomic cluster, the Pt single-atom and the Pt nanoparticle, and the fundamental research work reveals that the Pt-O-Pt type of structure is the more suitable catalytic moiety for the targeted reaction. While the work is potentially suitable for publication by Nature Communications, I feel the rigor needs being improved in several key aspects of the paper.

Response: We would like to thank the reviewer for the valuable comment. We have carefully revised the Manuscript and Supplementary Information according to the reviewer's comments. The revised content has been underlined and highlighted in yellow in the revised Manuscript and Supplementary Information. The point-by-point responses are shown in detail below.

Comment 1: *The computationally proposed Pt-O-Pt atomic structure is very particular in terms of its own structure and how the Co anchoring sites are located relative to the platinum species. Such a kind of precision and uniformity does not exist according to the HAADF-STEM images. Perhaps multiple plausible computational models should be proposed and examined to show the superiority of the Pt-O-Pt atomic cluster over the single-atom and nanoparticle counterparts are generally applicable.*

Response 1: We would like to thank the reviewer for the valuable comment. The specific Pt-O-Pt atomic structure is proposed with optimization based on the characterization results including HAADF-STEM, XPS, XANES *etc.* The magnified image of HAADF-STEM marked in Fig. 1c reveals that Co atoms are scattered around Pt ACs (dots with different contrasts, Fig. 1d), indicating the possible codependent relationship and interactions between Pt ACs and Co SAs. This has been verified by the high-resolution Co 2p spectrum of Pt-ACs/CoNC which slightly shifts compared to the pure CoNC in Fig. 2a. The result suggests the occurrence of electron transfer between Pt and Co atoms, confirming the existence of strong interaction between these two species. The XANES results in Supplementary Fig. 12a show slightly different white-line intensities in Co K-edge of Pt-ACs/CoNC compared to CoNC, further demonstrating the Pt incorporation slightly alters the charge density around Co atoms. Thus it can be concluded that there are strong interactions between Pt and Co atoms in Pt-ACs/CoNC. Furthermore, the EXAFS fitting results of Pt-ACs/CoNC (Fig. 2e) indicate the Pt atoms are separated by O atoms to form Pt-O-Pt units and each Pt atom is stabilized by three O atoms through Pt-O bonds in Pt-ACs/CoNC. Therefore, concluded from these characterization results, we proposed an optimized Pt₇O₁₂ cluster constructed on a CoN₄-carbon surface consisting of all the information mentioned above to be the most reasonable structure to simulate the atomic structure of Pt-ACs/CoNC.

Based on the reviewer's suggestion, we have proposed additional computational models of Pt-ACs/CoNC by varying the dimensions of Pt ACs and numbers of Co SAs to better illustrate our findings. To further avoid being too particular, we have constructed $\text{Pt}_{11}\text{O}_{16}$ and $\text{Pt}_{18}\text{O}_{24}$ (Pt bonding to O atoms with three coordination numbers) on top of one single Co atom as the anchoring site to better understand how such Pt ACs bind to Co atoms and the role of oxygen atoms in facilitating such an interaction. We have performed *ab initio* molecular dynamics (AIMD) to determine the structural configurations and stability of Pt ACs. As such, our simulations yield plausible structures with no bias involved. At the end of AIMD, the most stable structural configurations obtained are shown in Fig. R1. We find that the Pt-O-Pt atomic clusters are clearly supported by Co single atoms. Additionally, our simulations reveal that the Pt ACs are bonded to the Co atom *via* an O atom in each case, further demonstrating that O is essential for the attachment of the Pt ACs to the Co SAs.

Fig. R1 The atomic structure of (a) $\text{Pt}_{11}\text{O}_{16}$ and (b) $\text{Pt}_{18}\text{O}_{24}$ supported by Co single atoms.

As a next step, in addition to the HER free energy analysis performed during the initial submission, we have further investigated the HER catalytic ability of the aforementioned atomic structures of Pt-ACs/CoNC (shown in Fig. R1). We randomly chose five Pt sites (marked as yellow spheres) per structure and evaluated ΔG_H for the H adsorption with the same computational parameters used for the structure in the manuscript. The results show the best optimized free energies are -0.03 eV and -0.18 eV for the Pt atoms in $\text{Pt}_{11}\text{O}_{16}$ and $\text{Pt}_{18}\text{O}_{24}$, respectively, indicating that they are potentially HER active.

In conclusion, by performing additional structural studies as requested by the reviewer, we have demonstrated that, with the variation of either the cluster size ($\text{Pt}_{11}\text{O}_{16}$ and $\text{Pt}_{18}\text{O}_{24}$ supported by one Co single atom, Fig. R1) or number of anchoring Co sites (Pt_7O_{12} on three Co single atoms, Fig. 4 in the manuscript), all of the resulting structures can potentially deliver excellent HER catalytic activity, verifying the advantages of Pt-O-Pt in Pt ACs for HER process.

We have added the atomic structures of $\text{Pt}_{11}\text{O}_{16}$ and $\text{Pt}_{18}\text{O}_{24}$ supported by Co single atoms in Supplementary Fig. 25 on Page 31. The corresponding discussion has been included on Page 15 in the revised Manuscript and Page 31 in the revised Supplementary Information. We have updated the DFT methods section in the Supplementary Information on Page 5 to include the AIMD methods.

Page 15 in the revised Manuscript: A Pt₇O₁₂ cluster was constructed on CoN₄-carbon surface to simulate the atomic structure of Pt-ACs/CoNC (Fig. 4a, the other constructed atomic structures are shown in Supplementary Fig. 25).

Page 31 in the revised Supplementary Information: In addition to the atomic structure we have constructed in Fig. 4 in the manuscript, we have proposed additional computational models of Pt-ACs/CoNC by varying the dimensions of Pt ACs and numbers of Co SAs to better illustrate our findings. To further avoid being too particular, we have constructed Pt₁₁O₁₆ and Pt₁₈O₂₄ (Pt bonding to O atoms with three coordination numbers) on top of one single Co atom as the anchoring site. After relaxing these configurations by *ab initio* molecular dynamics (AIMD), the most stable structural configurations obtained are shown in Supplementary Fig. 25a-b. We find that the Pt-O-Pt atomic clusters are clearly supported by Co single atoms. Additionally, our simulations reveal that the Pt ACs are bonded to the Co atom *via* an O atom in each case, further demonstrating that O is essential for the attachment of the Pt ACs to the Co SAs. We have further investigated the HER catalytic ability of the aforementioned atomic structures of Pt-ACs/CoNC. We randomly chose five Pt sites (marked as yellow spheres) per structure and evaluated ΔG_H for the H adsorption. The results show the best optimized free energies are -0.03 eV and -0.18 eV for the Pt atoms in Pt₁₁O₁₆ and Pt₁₈O₂₄, respectively, indicating that they are potentially HER active. Therefore, changing the numbers of Pt and Co atoms in Pt-ACs/CoNC (DFT calculations) can all deliver excellent HER catalytic activity, suggesting the advantages of Pt ACs for HER process.

Page 5 in the revised Supplementary Information: Ab initio molecular dynamics (AIMD) simulations were also carried out using VASP. The wavefunctions were expanded with a kinetic energy cut-off value of 450 eV. Otherwise, similar parameters mentioned above were employed. MD simulations were carried out using the NVT ensemble using a Nose-Hoover thermostat, and the temperature was ramped up from 10 K to 310 K over a time period of 4 ps. A time-step of 1 fs was used in all our simulations.

We have also generated additional atomic structures, i.e. Pt₁₁O₁₆ and Pt₁₈O₂₄, on a single CoN₄ moiety embedded in a basal carbon plane to explore alternative plausible AC structures. The stability of these ACs is tested using AIMD simulations.

Comment 2: *I am a bit concerned if the present CoNC platform is suitable for the general interrogation of single-atom Pt vs. Pt-O-Pt catalytic center? To answer such a type of fundamental question, usually researchers would choose the uniform or unbiased support/substrate system to carry the supported catalytic centers to conduct the investigation. However, in this work, the extent of interaction between the platinum atoms and the cobalt atoms are bound to be different as the function of platinum structures/sizes, so the varied catalytic performance is no longer a sole function of interatomic Pt-O-Pt interactions. The role of Co should be further understood in this work. Notably, from Ref. 15, the single-atom Pt catalyst seems even more active than the best catalyst reported in this work. This further manifest that the cobalt is having a*

role to play. The least to say is the role of Co in modifying Pt activity should be further understood.

Response 2: We would like to thank the reviewer for the valuable comment. We agree with the reviewer that it is usually necessary to choose the uniform or unbiased supports/substrates for the unaffected investigations. We have previously conducted a comparison experiment by immobilizing Pt species (similar Pt loading to Pt ACs) directly on N-doped carbon (NC) without the presence of isolated Co atoms. As shown in Supplementary Fig. 15, numerous Pt NPs, rather than Pt ACs, are observed on the NC matrix, demonstrating the crucial role of Co SAs played in directing the formation of Pt ACs. Thus to avoid such bias, we have chosen the same substrate, isolated Co atoms on N co-doped porous carbon (CoNC), as the support for all three species: Pt single atoms (SAs), Pt atomic clusters (ACs) and Pt nanoparticles (NPs) (highlighted on Page 2 and 6 in the revised Manuscript). This can efficiently eliminate the possible influence of the substrate on the supported structures.

The anchoring of Pt on Co atoms in CoNC can ensure the stable transition from Pt SAs gradually to Pt ACs when the incorporated Pt content increases. The strong interactions formed between Co and Pt atoms can effectively prevent the unwanted aggregation of Pt atoms into nanoparticles during the preparation and catalytic process. Furthermore, the strong electron transfer between the Co and Pt atoms also helps to stabilize the structure. In order to prove this, we have conducted additional calculations regarding the relationship between Co anchoring sites and Pt ACs, by rotating the structure of Pt ACs on CoNC to reveal potential bonding situation with C and N atoms (Fig. R2a-b). However, after optimizing, the Pt ACs always rotate back to form Pt-O-Co bonds with Co SAs on CoNC (Fig. R2c), further demonstrating the existence of Co SAs on CoNC is beneficial to stabilize Pt ACs.

Fig. R2 The top and side view of the constructed atomic structure of Pt-ACs/CoNC (a) Pt atoms bind with N atoms, (b) Pt atoms bind with C atoms, (c) Pt atoms bind with O atoms on top of Co.

As the same substrate (CoNC) has been chosen for anchoring Pt SAs and Pt ACs, the difference for Pt-SAs/CoNC and Pt-ACs/CoNC is the anchored Pt species, which are directly responsible for their HER catalytic capability. Moreover, the Pt atoms in Pt-

ACs/CoNC and Pt-SAs/CoNC share a similar coordination environment, where Pt atoms are stabilized by three O atoms and a Co-O-Pt interaction is established between the Pt species and CoNC substate (EXAFS fitting results in Fig. 2 and Supplementary Fig. 14). Therefore, the major difference between these two candidates is that Pt-ACs/CoNC possess the Pt-O-Pt units. Thus, it can be summarized that these units contribute significantly to the superior HER catalytic activity. The DFT calculations also confirm that the charge density distributions of Pt-O-Pt species in Pt-ACs/CoNC reveal less charge delocalized from Pt to the nearby O atoms, making these Pt atoms with higher electron density, accounting for their superior HER catalytic activity. The Co-O-Pt interface in Pt-SAs/CoNC and Pt-ACs/CoNC can help to stabilize Pt species on CoNC, but the electron transfer from Co/Pt atoms to O atoms results in their relatively reduced electron density compared to the Pt atoms in Pt-O-Pt units. Therefore, Pt-SAs/CoNC exhibits lower HER catalytic activity compared to Pt-ACs/CoNC.

Nevertheless, both Co and Pt transfer electron to O atoms in Pt-O-Co interface can suppress the electron from Pt to O to some extent compared to the one without Co anchoring site. Therefore, the bonding of Pt-O-Co can still benefit the electron structure of Pt ACs. Through the electron transfer *via* the Pt-O-Co bonds (Fig. 4b), the electron distributions on Pt ACs are modulated to create an optimized environment for H adsorption, leading to increase HER performance.

We appreciate that the reviewer has suggested Ref 15 for comparison. We agree with the reviewer that the support effect for enhancing the HER activity of the anchored Pt species is very important, and single Co atoms indeed play important roles beyond just anchoring sites (see discussion above). In Ref 15, the authors reported the anchoring of atomically dispersed Pt on WO_{3-x} as substrate (Pt SA/m- WO_{3-x}) to enhance the HER performance. The catalysts exhibited excellent HER activity with overpotentials of ~ 50 mV at current densities of 10 mA cm^{-2} . The loading of Pt on Pt SA/m- WO_{3-x} was 0.42 wt %. The mass activity of Pt SA/m- WO_{3-x} was 12.8 A mg^{-1} at the overpotential of 50 mV. In our manuscript, we designed Pt ACs on top of CoNC as substrate, and achieved 10 mA cm^{-2} at the overpotential at 24 mV. The Pt loading in Pt-ACs/CoNC was 0.52 wt% and the mass activity was 28.6 A mg^{-1} at the overpotential of 50 mV, which is higher than that of Pt SA/m- WO_{3-x} . Furthermore, our reported Pt-SAs/CoNC also showed comparable performance comparing to the report (10 mA cm^{-2} at an overpotential of 57 mV and mass activity is $18.8 \text{ A mg}^{-1}_{\text{Pt}}$ at 50 mV). This is in agreement with the reviewer's comments, and the supporting effect is crucial for achieving enhanced HER performance. Compared to Ref 15, we used different substrate (CoNC), and single Co atoms effectively modulate the electron structures of the Pt atoms *via* forming covalent bonds. Furthermore, the Co anchoring sites can also facilitate and stabilize Pt ACs with Pt-O-Pt units, which is highly beneficial for HER process.

Comment 3: *The synthesis, if without further justification for the Pt precursor impregnation, is somewhat lacking elegance to truly form uniform structures with atomic precision. For example, how do we make sure the Co terminal can attract all the Pt forming exclusive types of moieties as single atom, atomic cluster, or*

nanoparticles? What is the surface bonding environment that promotes the exclusive formation of either one of the three representative species?

Response 3: We would like to thank the reviewer for the valuable comment. The Pt-ACs/CoNC composite is synthesized by anchoring the atomic-layered Pt ACs to Co SAs on a highly porous CoNC substrate *via* a simple wet-impregnation process. The characterization results of CoNC substrate in Supplementary Fig. 3 reveal that the isolated Co atoms are positively charged and stabilized by four N atoms in the equatorial plane and one O atom in the axial position. The bonds between Co and O lead to electrons transfer from Co SAs to the absorbed O atoms, resulting in higher electron density of these O atoms and making them easier to capture the incorporated Pt ions. The different types of Pt species, including single atom/atomic cluster/nanoparticle, are proceeded *via* a gradual formation process when different amounts of Pt ions are incorporated. Initially, when Pt ions are added into CoNC, they tend to attach to O atoms on Co SAs, forming the Pt SAs protrusive structure. At this stage, the Pt ions will occupy the Co-O species and form homogeneous distribution at low Pt content. With the increase of Pt loading, some of the Pt species will be captured and attached on Pt SAs protrusive structure, leading to the formation of Pt ACs supported by the isolated Co atoms (*Nat. Commun.* 2017, 8, 1070; *Nat. Commun.* 2019, 10, 3808). The content of Pt ACs continuously increases until most Pt SAs sites are occupied. Further increasing the Pt loading, more Pt atoms will be attached on the Pt ACs, turning the Pt ACs into the Pt NPs and reducing the utilization of Pt atoms. The electrochemical performance of Pt SAs, Pt ACs, and Pt NPs has a trend of a volcano plot, that the gradual transition from Pt SAs to Pt ACs leads to the increase of electrochemical performance while the change from Pt ACs to Pt NPs degrades the performance. This is consistent with the proposed gradual formation mechanism.

In order to better understand the role of O in facilitating Pt binding to CoNC units, we performed an additional AIMD simulation by randomly placing O and Pt atoms in the vicinity of the CoNC unit and allowing the system to equilibrate (see DFT methods in the revised Supplementary Information). Our results reveal three possible metastable configurations of Pt SAs on CoNC, which are shown in Fig. R3 (named as Pt-SAs/CoNC-A, Pt-SAs/CoNC-B and Pt-SAs/CoNC-C), obtained at different time intervals during the AIMD run. Further DFT relaxations indicated that the Pt SAs/CoNC-C is the most stable of these three structures, which is consistent with the model we proposed for Pt-SAs/CoNC (with the adsorption of O₂ molecules, Supplementary Fig. 14). In this configuration, the O atom acts as an anchoring link for Pt species to bind to the CoN₄ unit. Given that this is the most stable configuration, it clearly highlights the critical role played by the O atom as an anchor. This further highlights that the O atoms above Co SAs can capture the incorporated Pt ions forming Pt-SAs, and then forming Pt ACs and Pt NPs.

Fig. R3 Three different metastable configurations of (a) Pt-SAs/CoNC-A, (b) Pt-SAs/CoNC-B and (c) Pt-SAs/CoNC-C.

We have added these results in Supplementary Fig. 3e and the corresponding explanation on Page 9 in the revised Supplementary Information.

Page 9 in the revised Supplementary Information: In order to understand the role of O in facilitating Pt binding to CoNC units, we performed an additional AIMD simulation by randomly placing O and Pt atoms in the vicinity of the CoNC unit and allowing the system to equilibrate (see DFT methods). Our results reveal three possible metastable configurations of Pt SAs on CoNC, which are shown in Supplementary Fig. 3e (named as Pt-SAs/CoNC-A, Pt-SAs/CoNC-B and Pt-SAs/CoNC-C), obtained at different time intervals during the AIMD run. Further DFT relaxations indicate that the Pt-SAs/CoNC-C is the most stable of these three structures. In this configuration, the O atom acts as an anchoring link for Pt species to bind to the CoN₄ unit. Given that this is the most stable configuration, it clearly highlights the critical role played by the O atom as an anchor to capture the incorporated Pt ions forming protrusive atomic structure.

Comment 4: *If the formation of increasingly aggregated platinum species is a sole function of higher platinum loading, why the Pt-NPs/CoNC sample lacks the attributes of the Pt-ACs/CoNC sample according to HER activity data and the Tafel slope? Even if the Pt NPs are having zero activity for the given testing conditions, one would expect the remainder of the Pt ACs in the same sample to perform equally well as in the Pt-ACs/CoNC sample.*

Response 4: We would like to thank the reviewer for the valuable comment. The lower HER catalytic activity of Pt-NPs/CoNC compared to that of Pt-ACs/CoNC is ascribed to the aggregated Pt species in Pt-NPs/CoNC composite that has limited the accessible active sites, reducing the Pt utilization especially under the condition of low Pt content (around 1.47 wt%), which significantly reduces the overall activity of the catalyst.

As the response to Comment 3, the formation of Pt NPs relies on the sacrifice of Pt ACs, which means when further increasing the Pt loading, Pt atoms will be attached on the as-formed Pt ACs, turning these Pt ACs into Pt NPs. Therefore, Pt NPs are the dominant Pt moieties in Pt-NPs/CoNC, as demonstrated in the HAADF-STEM images and elemental mapping results in Supplementary Fig. 8 (almost no Pt ACs are observed in the Pt-NPs/CoNC sample). Therefore, even though the Pt loading in Pt-NPs/CoNC

(1.47 wt%) is higher than those in Pt-ACs/CoNC (0.52 wt%), the lower exposed Pt atoms of Pt-NPs/CoNC leads to the relatively lower HER catalytic activity compared to that of Pt-ACs/CoNC.

Comment 5: *Some explanations are required for the noticeable shifts of Co peaks in the XPS data reported in Fig. 2. The authors also claimed that supplementary Figure 3 reveal that isolated Co atoms are negatively charge. How? Also, the fitting of XPS data of core level Co-2p needs reconsideration. “The higher electron density of these O atoms makes them easier to capture the incorporated Pt ions” line-111-114. The above sentence needs further explanation and proof.*

Response 5: We would like to thank the reviewer for the valuable comment. The high-resolution XPS Co 2p spectrum of CoNC in Fig. 2a shows the peaks at 780.54 and 795.97 eV, revealing that Co species are at their oxidation states (the high-resolution Co 2p spectrum of metallic Co should be at 794.7 and 779.2 eV, *J. Am. Chem. Soc.* 2019, 141, 20118-20126), further demonstrating their isolated properties as SAs. Intriguingly, slight shifts of these peaks are spotted in Pt-SAs/CoNC and Pt-ACs/CoNC (more obvious), suggesting the Pt species interacted with Co SAs on the carbon substrate, which demonstrate the incorporation of Pt to CoNC exhibits an obvious electronic modulation of Co atoms and form strong metal-support interaction effect. Moreover, the charge density distributions in Fig. 4b also show the electron transfer occurred at the Co-O-Pt interface, resulting in strong interaction between Pt ACs and CoNC substrate, which is in consistent with the binding shift in high-resolution Co 2p spectra (Fig. 2a).

We have provided more explanations for the shifts of Co peaks in the revised Manuscript on Page 7-8.

Page 7-8 in the revised Manuscript: “The high-resolution Co 2p spectrum of CoNC in Fig. 2a shows the peaks at 780.54 and 795.97 eV, revealing that Co species are at their oxidation states, further demonstrating their isolated properties as SAs.⁹ Intriguingly, slight shifts of these peaks are spotted in Pt-SAs/CoNC and Pt-ACs/CoNC (more obvious), suggesting that the Pt species interacted with Co SAs on the carbon substrate, which demonstrate the incorporation of atomically dispersed Pt species to CoNC exhibits an obvious electronic modulation of Co atoms and establishes a strong metal-support interaction.”

XPS and XAS analyses of CoNC in Supplementary Fig. 3 reveal the isolated Co atoms are positively charged (we have revised the “negatively charged” to “positively charged”). The high-resolution Co 2p spectrum of CoNC in Supplementary Fig. 3a shows the peaks at 780.54 and 795.97 eV, revealing that Co species are at their oxidation states (the high-resolution Co 2p spectrum of metallic Co should be at 794.7 and 779.2 eV) (*J. Am. Chem. Soc.* 2019, 141, 20118-20126; *Nat. Energy* 2016, 1, 15006). The X-ray absorption near edge structure (XANES) results of Co K-edge in Supplementary Fig. 3b show the absorption edges of CoNC locates between Co foil and Co₃O₄,

indicating that the valence states of Co species in CoNC is between 0 and +2 (*J. Am. Chem. Soc.* 2019, 141, 20118-20126). Therefore, the XPS and XANES results of CoNC in Supplementary Fig. 3 reveal the isolated Co atoms are positively charged.

We have revised the corresponding sections accordingly in the revised Manuscript on Page 4-5.

Page 4-5 in the revised Manuscript: “X-ray photoelectron spectroscopy (XPS) and XAS analyses in Supplementary Fig. 3a-d reveal the isolated Co atoms are positively charged and stabilized by four surrounding N atoms in the equatorial plane and one O atom in the axial position.”

The high-resolution of Co 2p in Fig. 2a mainly shows the oxidation state of Co SAs and the binding shift after the incorporation of Pt species. As we have discussed above, the oxidation state of Co in Pt-SAs/CoNC, Pt-ACs/CoNC and Pt-NPs/CoNC are at their oxidation states. The slight shifts of Co 2p peaks for Pt-SAs/CoNC and Pt-ACs/CoNC (more obvious) compared to CoNC indicate the interactions between incorporated Pt species and Co SAs on the carbon substrate. Based on the reviewer’s suggestion, we have re-done the fitting of XPS data to better demonstrate our illustration (Fig. R4).

We have revised the fitting of XPS data in Fig. 2 in the revised Manuscript.

Fig. R4 The high-resolution XPS spectra of Co 2p in CoNC, Pt-SAs/CoNC, Pt-ACs/CoNC and Pt-NPs/CoNC.

The EXAFS fitting results of CoNC in Supplementary Fig. 3d show the Co SAs are stabilized by four N atoms in the equatorial plane and one O atom in the axial position, making Co SAs positively charged (XPS and XANES result of Co in Fig. 2a and Supplementary Fig. 3). The bonds between Co and O atoms lead electrons transfer from Co to O atoms and these O atoms with higher electron density are beneficial to capture the incorporated Pt ions. In order to further illustrate the fact that O-terminals on CoNC are an advantage to adsorb Pt ions, we have obtained different Pt-SAs/CoNC models by incorporating Pt ions on top of Co atoms (as described earlier in response to Comment 3). The results show three different metastable configurations have been achieved, as reproduced in Fig. R5 (named as Pt-SAs/CoNC-A, Pt-SAs/CoNC-B and Pt-SAs/CoNC-C). Further total-energy DFT calculations were performed to determine

the most stable structure. These calculations showed that Pt-SAs/CoNC-C is the most stable structure, where the Pt single atom is bonded to the Co atom *via* an O atom. This further highlights that the O atoms above Co SAs can capture the incorporated Pt ions forming the Pt-SAs, and then Pt ACs and Pt NPs. This result echoes our demonstration in the manuscript “*The higher electron density of these O atoms makes them easier to capture the incorporated Pt ions*”.

Fig. R5 Three different metastable configurations of (a) Pt-SAs/CoNC-A, (b) Pt-SAs/CoNC-B and (c) Pt-SAs/CoNC-C.

We have added the DFT results in Supplementary Fig. 3e, the corresponding explanation on Page 9 in the revised Supplementary Information and on Page 5 in the revised Manuscript.

Page 9 in the revised Supplementary Information: In order to understand the role of O in facilitating Pt binding to CoNC units, we performed an additional AIMD simulation by randomly placing O and Pt atoms in the vicinity of the CoNC unit and allowing the system to equilibrate (see DFT methods). Our results reveal three possible metastable configurations of Pt SAs on CoNC, which are shown in Supplementary Fig. 3e (named as Pt-SAs/CoNC-A, Pt-SAs/CoNC-B and Pt-SAs/CoNC-C), obtained at different time intervals during the AIMD run. Further DFT relaxations indicate that the Pt-SAs/CoNC-C is the most stable of these three structures. In this configuration, the O atom acts as an anchoring link for Pt species to bind to the CoN₄ unit. Given that this is the most stable configuration, it clearly highlights the critical role played by the O atom as an anchor to capture the incorporated Pt ions forming protrusive atomic structure.

Page 5 in the revised Manuscript: We have also revised “The higher electron density of these O atoms makes them easier to capture the incorporated Pt ions” to “These O atoms above Co single atoms can capture the incorporated Pt ions to form the protrusive atomic structure (Supplementary Fig. 3e)”.

Comment 6: Line-303-305: “*promising HER catalysts for next generation PEM electrolyzers*”. Since the study was done on GCE (< 3mm dia), this statement should be reserved until and unless authors have done- (1) large area study with catalysts deposited on gas diffusion electrodes. (2) The Methanol and CO tolerance study.

Response 6: We would like to thank the reviewer for the valuable comment. As the prepared Pt-ACs/CoNC displays excellent HER catalytic activity in the three-electrode system, we speculate it should be a promising candidate for next-generation PEM electrolyzers. Our next plan is to explore the application of Pt-ACs/CoNC in PEM electrolyzer, which could utilize the high atomic efficiency of Pt-ACs/CoNC to reduce the cost. However, this is out of our scope in this manuscript, and not our focus at the current stage. Therefore, we agree with the reviewer that until we have achieved some results in this area, we should refrain such claims in the manuscript.

Page 12 in the revised Manuscript: “With such a high activity and low Pt loading (0.52 wt%), Pt-ACs/CoNC is an active and cost-effective catalyst for hydrogen production in acidic media.”

Comment 7: *Authors should mention in the caption the details of chronoamperometry as well as the accelerated LSV cycle study. At what voltage it was performed. What is the potential window for LSV cycling? At what voltage the EIS study was done?*

Response 7: We would like to thank the reviewer for the valuable comment. We have added the required details in the revised Manuscript on Page 11 and Supplementary Information on Page 4 and 26.

Page 4 and 25 in the revised Supplementary Information: “The cycling performance of Pt-ACs/CoNC based electrodes was tested by repeating linear sweep voltammograms (LSV) running for 5000 cycles (accelerated scan rate 100 mV s^{-1}) with the potential range between 0.10 and -0.15 V vs. RHE.”

Page 11 in the revised Manuscript and Page 4 and 26 in the revised Supplementary Information: “The current-time plots were obtained at overpotentials of 27 mV and 63 mV to achieve the current density of 10 and 40 mA cm^{-2} respectively.”

Page 11 in the revised Manuscript and Page 4 in the revised Supplementary Information: “The electrochemical impedance spectra (EIS) were recorded with a frequency ranging from 10^6 to 0.01 Hz and an amplitude of 5 mV at an overpotential of 20 mV.”

Comment 8: *Considering the fact that HER is a reductive reaction and the platinum species, especially when being small, are very sensitive to air reoxidation. Many of the key characterization pertaining to the oxidation state of Pt-O-Pt should be carried out systematically in situ or with good sealing after reaction.*

Response 8: We would like to thank the reviewer for the valuable comment. We agree with the reviewer that air re-oxidation can potentially be a significant issue for the characterizations, especially for the metallic Pt with small sizes. However, in our study, the Pt species in Pt ACs are already coordinated with three O atoms (relatively stable), therefore it is very difficult for them to be fully reduced or further oxidized (*J. Am. Chem. Soc.* 2020, 142, 5594-5601; *Energy Environ. Sci.* 2017, 10, 2450; *Nat. Energy* 2019, 4, 512-518). When we performed the experiments, we were still very

careful to store the freshly prepared samples and the samples after the stability test to avoid any possible oxidation in the air. After the stability test, we have washed, collected and stored the samples in the vacuumed desiccator to avoid the reoxidation of Pt species if exposed to air. We have performed XPS characterization to detect the possible change of Pt species before and after the stability test. The high-resolution XPS results of Pt 4f in Supplementary Fig. 21c show the Pt species still possess cationic nature after the stability test, indicating the Pt ACs were well maintained. This is consistent with the HAADF-STEM results in Supplementary Fig. 21a-b (no aggregation nanoparticles). Moreover, the slightly negative shift of Pt 4f peaks after the stability test compared to the original samples suggests the Pt species are slightly reduced during HER process (reduction reaction).

We have added these operation procedures in the experimental sections of the revised Supplementary Information on Page 4 to better address the experimental processes.

Page 4 in the revised Supplementary Information: “After the stability test, the samples were washed, collected and stored in the vacuumed desiccator to avoid the reoxidation of Pt species”.

Reviewer #2:

In this manuscript, Zhao et al. use a combination of various experimental methods and computation to evaluate Pt-O-Pt clusters bound to CoN₄C sites on graphitic carbon that they have synthesized as well as Pt nanoparticles on CoN₄Co and Pt single atom catalysts supported on CoN₄Co and Pt supported on C supports. These results represent an important advance and will be of significance to the electrocatalysis field. Although I was impressed with the advances made in this MS, it also suffers from a number of flaws that must be addressed before it is publishable in a journal of the quality of Nat. Comm. There are several technical questions that should be addressed, and the writing needs considerable improvement.

Response: We would like to thank the reviewer for the valuable comment. We have carefully revised the Manuscript and Supplementary Information according to the reviewer's comments. The revised content has been underlined and highlighted in yellow in the revised Manuscript and Supplementary Information. The point-by-point responses are shown in detail below.

Strengths:

Comment 1: *The improved HER specific activity the authors were able to achieve with the "atomic cluster" catalysts compared to both SA and NP catalysts (Fig 3b). There is clearly something unique about these active sites beyond just being monodisperse.*

Suggestion: Some sort of error analysis (like error bars showing standard deviations) would make this aspect even stronger.

Response 1: We would like to thank the reviewer for the valuable comment. As we have demonstrated in the manuscript, the Pt atomic clusters combine the high atomic utilization of atomically dispersed catalysts and unique atom-to-atom interactions of cluster catalysts. Based on the experimental and DFT results, the HER active sites of Pt-ACs/CoNC are Pt-O-Pt units, where both Pt and O atoms are responsible for adsorbing H⁺ from the electrolyte. The low valence state of Pt atoms in Pt-ACs/CoNC implies fewer electrons have been transferred to the nearby O atoms, leading to a lower electron density on these O linkers. It significantly reduces the capability of O atoms to form stable onium cations with H⁺. Therefore, the adsorbed H⁺ can be easily desorbed from the O atoms and transferred to the surrounding Pt atoms. Furthermore, the higher electron density of Pt atoms in Pt-O-Pt units (XPS and XAS results) make them highly favourable for both H⁺ adsorption and HER process. The above merits contribute to the excellent HER catalytic activity of Pt-ACs/CoNC.

In contrast, the aggregation of Pt NPs in Pt-NPs/CoNC significantly reduces the Pt utilization, especially under the condition of low Pt content (around 1.5 wt%), which significantly reduces the overall HER activity of the catalyst compared to Pt ACs. The Pt atoms are sparsely distributed in Pt-SAs/CoNC, which has rendered them far apart from each other. As a consequence, the interactions between Pt atoms are negligible. Moreover, based on the DFT calculations, heavier electron transfer occurred in the Co-O-Pt interface (compared to the Pt-O-Pt units in Pt-ACs/CoNC), resulting in reduced electron density around these Co/Pt atoms, thus inducing the relatively lower HER

activity of Pt-SAs/CoNC compared with Pt-ACs/CoNC. In conclusion, Pt-ACs/CoNC displayed superior HER catalytic activity compared to those of Pt-SAs/CoNC and Pt-NPs/CoNC at the same test conditions is mainly ascribed to the advantages of atomically dispersed properties, adjacent atoms, low valence state, and O linkers in Pt-O-Pt units.

During the experiments, we have repeated HER test at the same condition several times to confirm the repeatability of the performance. Based on the reviewer's suggestion, we have added the error analysis with these results in Fig. R1. The results with error analysis confirm the Pt-ACs/CoNC exhibits much superior HER catalytic activity compared to those of Pt-SAs/CoNC and Pt-NPs/CoNC.

Fig. R1 The overpotential of current density of 10 mA cm⁻² for Pt-SAs/CoNC, Pt-ACs/CoNC and Pt-NPs/CoNC with the error analysis.

We have added the error analysis in the Supplementary Fig. 16 on Page 22.

Comment 2: *The authors performed an extensive amount of characterization of active sites of the AC catalyst. This provides insight into why these ACs are more active than the SA or NP catalysts. The XANES results showing the lower Pt valence state for the atomic cluster catalysts are particularly interesting.*

Suggestion: The authors claim that Pt bound to O in the Pt-O-Pt configurations has a lower oxidation state than SA Pt catalysts. The reasoning for this is not clear as O bound to Pt usually raises the oxidation state of Pt.

Response 2: We would like to thank the reviewer for the valuable comment. Pt atoms are stabilized by O atoms in both Pt-SAs/CoNC and Pt-ACs/CoNC samples. As shown in Fig. 2 and Supplementary Fig. 14, the EXAFS fitting results of Pt-SAs/CoNC show one O atom between Co and Pt atoms, and one chemisorbed O₂ molecule at the terminal position, indicating Pt atoms in Pt-SAs/CoNC bonded with three O atoms. Similarly, the EXAFS fitting results of Pt-ACs/CoNC in Fig. 2e indicate the Pt atoms are separated by O atoms to resemble isolated Pt species by forming Pt-O-Pt units and supported on CoNC through Co-O-Pt, thus the Pt atoms in Pt-ACs/CoNC are also stabilized by three O atoms through Pt-O bonds.

As the reviewer is concerned, the Pt valence state indeed will raise when they bond with O atoms or other light atoms compared to the metallic Pt species. But in this manuscript, the lower Pt valence state means the valence state of Pt atoms in Pt-

ACs/CoNC is relatively lower specifically compared to that of Pt-SAs/CoNC. Especially in both two configurations, the Pt atoms are bonded with the same number of O atoms. The DFT calculation results (charge density distributions in Fig. 4b) show heavier electron transfer occurred in the Co-O-Pt interface (compared to the Pt-O-Pt units) in Pt-ACs/CoNC, resulting in reduced electron density around these Co/Pt atoms. The Co-O-Pt interface in Pt-SAs/CoNC is similar to the interface in Pt-ACs/CoNC, in which the Pt atoms show reduced electron density. In contrast, the charge density distributions of Pt-O-Pt species in Pt-ACs/CoNC reveal less charge delocalized from Pt to the nearby O atoms, making these Pt atoms with higher electron density, which is also consistent with the XPS and XANES results in Fig. 2 and Supplementary Fig. 11c. Based on the above explanation, even though Pt atoms in Pt-SAs/CoNC and Pt-ACs/CoNC display similar atomic configurations (Pt stabilized by three O atoms), the less charge delocalized from Pt to the nearby O atoms in Pt-O-Pt units makes the Pt-ACs/CoNC possess lower average valence state compared to that of Pt-SAs/CoNC.

Comment 3: *The overall advance and narrative describing the advance is simple and relatively clear. They authors synthesize dispersed Pt catalysts that are more active than other monodisperse catalysts due to specific interactions between the Pt atoms and CoNC support. These interactions are well-explored with experimental approaches and some supporting computational analysis is provided. The end result is a more efficient HER catalyst.*

Suggestion: Although the story is relatively clear, the manuscript suffers from numerous grammatical errors and awkward and confusing phrasing. This makes the paper unnecessarily difficult to read and lowers its overall quality.

Suggestion: Although the story is simple, to increase the impact of this work and make it appropriate for a Nature journal, it should provide some perspective into how this approach can have broader impact beyond HER by supported Pt catalysts.

Response 3: We would like to thank the reviewer for the valuable comment. We have revised the entire manuscript very carefully, and have corrected all the grammatical errors and confusing phrasing to make the revised Manuscript more readable and smoother.

According to the reviewer's suggestion, we have added the perspective at the end of the conclusion part on Page 18 in the revised Manuscript.

Page 18 in the revised Manuscript: Our studies highlight the significance of creating Pt-O-Pt atomic clusters supported on atomically dispersed Co atoms for boosting the HER process. These findings also shed light on the designed synthesis of high-performance catalysts with delicately controlled active centers. Moreover, the constructed unique catalytic system may serve as a powerful platform for future studies of many other energy conversion reactions and organic synthesis, such as oxygen reduction reaction, methanol/alcohol oxidation reaction, n-butane dehydrogenation reaction, to name a few.

Weaknesses:

Comment 4: *Figs 3a and 3c show the AC and generic Pt/C catalyst performed almost identically. There is no indication of experimental error, so it's hard to believe that these are significantly different. Is this due to them reaching some sort of "maximum rate" for their experimental system (e.g., being transport limited?) or is it just a coincidence? Are there conditions/catalyst loadings where they can show a significant separation between performance of these two catalysts? Furthermore, the specific activity will be affected if they are reaching a systematic maximum rate.*

Suggestion: The claim that the Pt-AC catalyst performs better needs to be supported by some sort of error analysis.

Response 4: We would like to thank the reviewer for the valuable comment. The data shown in Fig 3a and 3c are obtained with the same total mass loading of $262 \mu\text{g cm}^{-2}$ for Pt-ACs/CoNC and commercial Pt/C (in which the element Pt loading is 1.31 and $52.4 \mu\text{g cm}^{-2}$ for Pt-ACs/CoNC and commercial Pt/C, respectively). Generally, commercial Pt/C is often considered as a benchmark catalyst for HER, as commercial Pt/C usually exhibits excellent catalytic activity for HER (*Nat. Commun.*, 2014, 5, 3783). Our results show that both Pt-ACs/CoNC and commercial Pt/C display similar HER catalytic activity, suggesting both of them are excellent for hydrogen generation. However, when enlarging the LSV results in Fig. 3a in the Manuscript, the as-prepared Pt-ACs/CoNC exhibit slightly superior HER catalytic activity, achieving a j of 10 mA cm^{-2} at the overpotential of 24 mV (26 mV at 10 mA cm^{-2} for commercial Pt/C), demonstrating the excellent capability of Pt-ACs/CoNC for HER.

The catalysts loading on the working electrode indeed influences the HER catalytic activity. We have tested the HER catalytic activity of commercial Pt/C with different mass loading. As shown in Supplementary Fig. 17, the HER activity of Pt/C decreases slightly when the Pt loading is lowered from 52.4 to $3.49 \mu\text{g cm}^{-2}$, and further reducing the Pt loading (to 2.62 and $1.36 \mu\text{g cm}^{-2}$) will lead to a rapid drop in the HER activity.

We have compared the Pt-ACs/CoNC and commercial Pt/C with the same total mass loading (Fig. 3a in the Manuscript) and same Pt loading (Fig. R2a). As shown in the manuscript (Fig. 3) and demonstrated in this response, Pt-ACs/CoNC shows slightly better HER catalytic activity than Pt/C with the same total mass loading. However, Pt-ACs/CoNC display much superior HER catalytic activity compared to commercial Pt/C with the same element Pt loading (Pt loading is $1.31 \mu\text{g cm}^{-2}$ for both samples), indicating the intrinsic catalytic activity of Pt ACs is superior compared to commercial Pt/C (Fig. R2a).

In the manuscript, in order to keep the same test conditions, we have chosen the same total loading of Pt-ACs/CoNC and commercial Pt/C on the working electrode. The mass activity of Pt-ACs/CoNC is 40 times higher than the Pt/C catalysts ($0.7 \text{ A mg}^{-1}_{\text{Pt}}$, Fig. 3b) with the same total loading, and at least more than 6 times superior to the Pt/C catalysts with optimized mass activity ($4.5 \text{ A mg}^{-1}_{\text{Pt}}$, Supplementary Fig. 17) (considering the mass activity will be affected if they are reaching a systematic maximum rate).

Similar to the response for Comment 1, we have performed HER test at the same

conditions for several times to confirm the repeatability of performance. Following the reviewer's suggestion, we have added the error analysis in Fig. R2b for Pt-ACs/CoNC and commercial Pt/C. The results show Pt-ACs/CoNC display better overpotential to achieve current density of 10 mA cm^{-2} with the error analysis.

Fig. R2 (a) The electrochemical performance test of Pt-ACs/CoNC and commercial Pt/C with same Pt loading on the working electrode, (b) the overpotential of current density of 10 mA cm^{-2} for Pt-ACs/CoNC and commercial Pt/C with the error analysis.

We have added the electrochemical performance test of Pt-ACs/CoNC and commercial Pt/C with the same element Pt loading on the working electrode (Pt loading is 1.31 μg cm^{-2} for both samples) and the overpotential of the current density of 10 mA cm^{-2} for Pt-ACs/CoNC and commercial Pt/C with the error analysis in Fig Supplementary Fig. 16 b-c on Page 22.

Comment 5: *The analysis of catalyst stability is limited to just the Pt-AC catalyst under one set of conditions where the catalyst is stable.*

Suggestion: It would be useful to know the conditions under which these active site structures become unstable, so as to direct future use of these materials.

Response 5: We would like to thank the reviewer for the valuable comment. Generally, the stability of an electrocatalyst should be assessed for long-term operation at certain overpotential or current, repeated LSV scans and post-characterizations after stability test. In our experiment, we have performed the stability test of Pt-ACs/CoNC at a current density of 10 mA cm^{-2} for 100 h, 40 mA cm^{-2} for 50 h, LSV repeating scans for 5000 cycles (Fig. 3f and Supplementary Fig. 20), and all the above tests show negligible loss of performance, demonstrating the exceptional electrochemical durability of the

Pt-ACs/CoNC in acidic HER process. Furthermore, we have performed post characterizations after the stability test. The HAADF-STEM and high-resolution TEM images in Supplementary Fig. 21a-b show the highly porous and graphitic structure has been well retained. The Pt ACs are also well maintained after the long-term HER process, as no nanoparticles are present in the images. The high-resolution XPS spectra of Pt 4f in Supplementary Fig. 21c show the Pt species after stability test still possess cationic nature, further indicating the Pt ACs are well maintained (no aggregation). The EIS results in Supplementary Fig. 21d show the spectra before and after the stability test remain almost the same, suggesting that the properties of Pt-ACs/CoNC are not altered during the long-term stability HER test. Moreover, the ICP-MS test for the collected H₂SO₄ solution after the long-term test shows a negligible amount of Pt and Co (< 3.43 and 99.4 ppb, respectively) dissolved in the solution. All these results demonstrate the high stability of the Pt-ACs/CoNC composite in catalyzing HER in the acidic environment.

Considering the reviewer's concern, the Pt-ACs/CoNC might be unstable under the extremely high currents (e.g. > 1 A cm⁻²) in proton exchange membrane (PEM) electrolyzers. The Pt ACs might be dissolved or the structure of carbon substrate might be destroyed under such high currents. In our future study, we will investigate the stability of the Pt-ACs/CoNC catalysts under harsh operational conditions in practical PEM electrolyzers.

Comment 6: *As mentioned above, the quality of the writing needs improvement. The grammar and phrasing are very poor throughout the ms (although less so in the computational section). This will greatly improve readability and avoid confusion.*

Suggestions (beyond a careful edit and proofread by someone fluent in English):

Use simpler terms for the different catalysts. E.g., Pt-ACs instead of Pt-ACs/CoNC or Pt-SA instead of for Pt-SAs/CoNC. The support is the same throughout, so no need to repeat it every time, especially because it makes the text cumbersome to read.

The Pt-SAs is not defined in the ms and no figure of the atomic structure is included for it, so the reader can only guess exactly what this structure is.

The names of the clusters are not very descriptive. SA, AC and NP essentially only indicate size, but not coordination. This is especially an issue because the SA structure is not defined. AC is particularly non-descript because it doesn't indicate that this isn't just a small cluster of Pt atoms. Maybe a name like Pt-O-C or Pt-O-AC is better.

Response 6: We would like to thank the reviewer for the valuable comment. We have polished the entire manuscript very carefully, including the grammar and phrasing.

We have kept the name as Pt-ACs/CoNC, Pt-SAs/CoNC for the catalysts as they can directly show their compositions, which is the Pt ACs or Pt SAs constructed on the CoNC substrate. Moreover, in order to keep the consistency with the definition of SAs and NPs, we have not changed ACs to Pt-O-C or Pt-O-AC.

We have defined the SAs as single atoms on Page 6 in the manuscript “By varying

the amount of Pt precursors in the synthetic procedure (see details in Methods), Pt single atoms (SAs) and nanoparticles (NPs) can be successfully obtained on the CoNC substrate with loadings of 0.17 and 1.47 wt%, as measured by ICP-MS, respectively.” Moreover, the HAADF-STEM images and the elemental mapping results of Pt-SAs/CoNC are shown in Supplementary Fig. 7. The HAADF-STEM image of Pt-SAs/CoNC in Supplementary Fig. 7a shows the highly porous structure and no particles or clusters were observed on the entire carbon matrix. Instead, the presence of numerous bright dots with different contrast is spotted. The observed dots with different contrast indicate the existence of two distinctive metal elements (in our case are Pt and Co atoms), together with other non-metal elements such as C, N and O (elemental mapping results in Supplementary Fig. 7c). Furthermore, the Pt and Co atoms are in close proximity to each other (as circled in Supplementary Fig. 7b), suggesting the formation of possible interactions between them. The EXAFS result of Pt-SAs/CoNC in Fig. 2e shows the absence of Pt-Pt contribution at ~ 2.5 Å, suggesting no Pt nanoparticles or Pt-Pt interactions are present in Pt-SAs/CoNC. Instead, the only dominant peak for Pt-SAs/CoNC locates around 1.65 Å, sharing a similar position to the typical exclusive Pt-O coordination environment in PtO₂, which demonstrates the Pt atoms in Pt-SAs/CoNC are coordinated by O atoms. The fitting results of Pt-SAs/CoNC (Supplementary Fig. 14) show one O atom between Co and Pt atoms, and one chemisorbed O₂ molecule at the terminal position.

Technical Issues:

Comment 7: A very specific Pt-O-AC structure was used for DFT modeling-do other similar models give similar or different spectra, H* binding energies, etc.

Response 7: Based on the reviewer’s suggestion, we have proposed additional computational models of Pt-ACs/CoNC by varying the dimensions of Pt ACs and numbers of Co SAs to better illustrate our findings. To further avoid being too particular, we have constructed Pt₁₁O₁₆ and Pt₁₈O₂₄ (Pt bonding to O atoms with three coordination numbers) on top of one single Co atom as the anchoring site to better understand how such Pt ACs bind to Co atoms and the role of oxygen atoms in facilitating such an interaction. We have performed *ab initio* molecular dynamics (AIMD) to determine the structural configurations and stability of Pt ACs. As such, our simulations yield plausible structures with no bias involved. At the end of AIMD, the most stable structural configurations obtained are shown in Fig. R3. We find that the Pt-O-Pt atomic clusters are clearly supported by Co single atoms (or in other words, bind to Co atoms). Additionally, our simulations reveal that the Pt ACs are bonded to the Co atom *via* an O atom in each case, further demonstrating that O is essential for the attachment of the Pt ACs to the Co SAs.

Fig. R3 The atomic structure of (a) $\text{Pt}_{11}\text{O}_{16}$ and (b) $\text{Pt}_{18}\text{O}_{24}$ supported by Co single atoms.

As a next step, in addition to the HER free energy analysis performed during the initial submission, we have further investigated the HER catalytic ability of the aforementioned atomic structures of Pt-ACs/CoNC (shown in Fig. R3). We randomly chose five Pt sites (marked as yellow spheres) per structure and evaluated ΔG_H for the H adsorption with the same computational parameters used for the structure in the manuscript. The results show the best optimized free energies are -0.03 eV and -0.18 eV for the Pt atoms in $\text{Pt}_{11}\text{O}_{16}$ and $\text{Pt}_{18}\text{O}_{24}$, respectively, indicating that they are potentially HER active.

In conclusion, by performing additional structural studies as requested by the reviewer, we have demonstrated that, with the variation of either the cluster size ($\text{Pt}_{11}\text{O}_{16}$ and $\text{Pt}_{18}\text{O}_{24}$ supported by one Co single atom, Fig. R3) or number of anchoring Co sites (Pt_7O_{12} on three Co single atoms, Fig. 4), all of the resulting structures can potentially deliver excellent HER catalytic activity, verifying the advantages of Pt-O-Pt in Pt ACs for HER process.

We have added the atomic structures of $\text{Pt}_{11}\text{O}_{16}$ and $\text{Pt}_{18}\text{O}_{24}$ supported by Co single atoms and corresponding H adsorption at different Pt sites in Supplementary Fig. 25 on Page 31. The corresponding discussion has been included on Page 15 in the revised Manuscript and Page 31 in the revised Supplementary Information. We have updated the DFT methods section in the Supplementary Information on Page 5 to include the AIMD methods.

Page 15 in the revised Manuscript: A Pt_7O_{12} cluster was constructed on CoN_4 -carbon surface to simulate the atomic structure of Pt-ACs/CoNC (Fig. 4a, the other constructed atomic structures are shown in Supplementary Fig. 25).

Page 31 in the revised Supplementary Information: In addition to the atomic structure we have constructed in Fig. 4 in the manuscript, we have proposed additional computational models of Pt-ACs/CoNC by varying the dimensions of Pt ACs and numbers of Co SAs to better illustrate our findings. To further avoid being too particular, we have constructed $\text{Pt}_{11}\text{O}_{16}$ and $\text{Pt}_{18}\text{O}_{24}$ (Pt bonding to O atoms with three coordination numbers) on top of one single Co atom as the anchoring site. After relaxing these configurations by *ab initio* molecular dynamics (AIMD), the most stable structural configurations obtained are shown in Supplementary Fig. 25a-b. We find that the Pt-O-Pt atomic clusters are clearly supported by Co single atoms. Additionally, our

simulations reveal that the Pt ACs are bonded to the Co atom via an O atom in each case, further demonstrating that O is essential for the attachment of the Pt ACs to the Co SAs. We have further investigated the HER catalytic ability of the aforementioned atomic structures of Pt-ACs/CoNC. We randomly chose five Pt sites (marked as yellow spheres) per structure and evaluated ΔG_H for the H adsorption. The results show the best optimized free energies are -0.03 eV and -0.18 eV for the Pt atoms in Pt₁₁O₁₆ and Pt₁₈O₂₄, respectively, indicating that they are potentially HER active. Therefore, changing the numbers of Pt and Co atoms in Pt-ACs/CoNC (DFT calculations) can all deliver excellent HER catalytic activity, suggesting the advantages of Pt ACs for HER process.

Page 5 in the revised Supplementary Information: Ab initio molecular dynamics (AIMD) simulations were also carried out using VASP. The wavefunctions were expanded with a kinetic energy cut-off value of 450 eV. Otherwise, similar parameters mentioned above were employed. MD simulations were carried out using the NVT ensemble using a Nose-Hoover thermostat, and the temperature was ramped up from 10 K to 310 K over a time period of 4 ps. A time-step of 1 fs was used in all our simulations.

We have also generated additional atomic structures, i.e. Pt₁₁O₁₆ and Pt₁₈O₂₄, on a single CoN₄ moiety embedded in a basal carbon plane to explore alternative plausible AC structures. The stability of these ACs is tested using AIMD simulations.

Comment 8: The DFT model is very simplified relative to the actual system. It does not include the effect of bias (e.g., using a grand canonical description that allows the electron number to change to match the Fermi level to the applied potential) nor solvent. There is no justification for why a vacuum model should accurately describe what is happening at the electrified interface.

Response 8: The effect of the bias, i.e. the applied potential, can be taken into account by the computational hydrogen electrode (CHE) scheme (*J. Phys. Chem. B* 2004, 108, 17886). The effect of potential in this case would be in simply changing the ΔG_H value linearly through an additive term, i.e. $\Delta G_H + V$ (where V is the applied potential). The applied potentials in our experiments for Pt ACs is about -0.05 to -0.1 eV. Our computations show that for Pt ACs, the ΔG_H values (barriers) also have similar absolute values. As such, it clearly shows that the barrier experienced through ΔG_H is negated by the application of the potential. Hence, in our experiments, the HER evolution occurs around -0.1 eV, where the current density is greatly enhanced.

As for the effect of solvent, this is a good question from the reviewer and ideally, it would be good to include these effects. However, it is a daunting task to include these effects within DFT. In fact, dedicated beyond-DFT codes that take into account the solvent effects are being developed (<https://jdftx.org/>, <https://github.com/henniggroup/VASPsol>). The effect of different solvents can in principle be taken in account using these advanced codes. However, they are within reach for simple organic solvents. Given that we have a combination of molecules and ions in our solvent, this is beyond the scope of this work. Hence, DFT studies tend to

neglect the effect of solvents when computing HER or oxygen evolution reaction (OER)/oxygen reduction reaction (ORR) potential energy diagrams, and rely on vacuum models.

Comment 9: No barriers are calculated so suggestions about the kinetics based on the DFT computed-thermodynamics (e.g., H^* binding free energies) is speculative.

Response 9: We would like to thank the reviewer for the valuable comment. As the HER process is fast, we have explored the HER catalytic mechanism of Pt-ACs/CoNC by DFT calculations (Volmer-Heyrovsky or Volmer-Tafel). In Volmer-Tafel pathway, two adsorbed hydrogen atoms on active sites combine to give H_2 . In contrast, in the Volmer-Heyrovsky pathway, the transfer of a second electron to the adsorbed hydrogen atom is coupled to the transfer of another proton from the solution to evolve H_2 . We have modeled both these pathways using DFT and the free energy diagram is shown in Fig. R4. The results show that the Volmer-Tafel pathway is more favorable since the reaction energy toward H_2 evolution is lower. Thus, Volmer-Tafel pathway is the dominant HER catalytic process of Pt-ACs/CoNC, which is consistent with our experimental findings. Instead, we find that there is relatively larger reaction energy for H_2 molecule to form at a Pt site of Pt-ACs/CoNC *via* the Volmer-Heyrovsky pathway, thus making it less probable in our experiments. Therefore, both of the DFT calculations and experimental results demonstrate the Volmer-Tafel pathway is more favorable for Pt-ACs/CoNC in HER process.

The calculation of the initial transition state, i.e. the transition state encountered during the $H^+ + e^-$ (ion in the solution) conversion to H^* (bound state) is not accessible *via* DFT calculations since modeling such isolated charges and their properties is not well-described within DFT. As such, DFT studies tend to neglect this transition state due to methodology limitations.

Fig. R4 Free energy diagrams of the Volmer-Heyrovsky and Volmer-Tafel catalytic pathways on Pt-ACs/CoNC.

We have added free energy diagrams of the Volmer-Heyrovsky and Volmer-Tafel catalytic pathways on Pt-ACs/CoNC in Supplementary Fig. 24 on Page 30 in the revised Supplementary Information, and the explanations on Page 14 in the revised Manuscript and Page 30 in the revised Supplementary Information.

Page 30 in the revised Supplementary Information: We have explored the HER catalytic mechanism of Pt-ACs/CoNC by using DFT calculations (Volmer-Heyrovsky or Volmer-Tafel). In Volmer-Tafel pathway, two adsorbed hydrogen atoms on active sites combine to give H₂. In contrast, in the Volmer-Heyrovsky pathway, the transfer of a second electron to the adsorbed hydrogen atom is coupled to the transfer of another proton from the solution to evolve H₂. We have modeled both these pathways using DFT and the free energy diagram is shown in Supplementary Fig. 24. The results show that the Volmer-Tafel pathway is more favorable since the reaction energy toward H₂ evolution is lower. Thus, Volmer-Tafel pathway is the dominant HER catalytic process of Pt-ACs/CoNC, which is consistent with our experimental findings. Instead, we find that there is relatively larger reaction energy for H₂ molecule to form at a Pt site of Pt-ACs/CoNC via the Volmer-Heyrovsky pathway, thus making it less probable in our experiments. Therefore, both of the DFT calculations and experimental results demonstrate the Volmer-Tafel pathway is more favorable for Pt-ACs/CoNC in HER process.

Comment 10: At what reducing potential are Pt-O-Pt structures reduced to Pt (or other degradation process)? How close is the operating bias to potentials at which degradation occurs?

Response 10: We would like to thank the reviewer for the valuable comment. Following the reviewer's suggestion, we tested the Pt-ACs/CoNC deposited on carbon paper under the constant potential at -0.5 V *vs.* RHE for 5 hours. We then collected the samples for XPS analysis. The high-resolution of Pt 4f XPS result in Fig. R5 shows the Pt species in Pt-ACs/CoNC after test still possess cationic nature (peaks at 75.8 and 72.5 eV), whereas the peaks for metallic Pt should be at 74.8 and 71.5 eV (*Nat. Catal.* 2018, 1, 985-992), further indicating the structure of Pt ACs is well maintained and Pt-O-Pt structures have not been reduced to Pt even at the very negative potential of -0.5 V *vs.* RHE. The potential range for LSV measurements of Pt-ACs/CoNC in the manuscript is between 0.10 V and -0.15 V *vs.* RHE, and the current-time plots were obtained at potentials of 27 mV and 63 mV *vs.* RHE, which is more positive than the -0.5 V. Therefore, the Pt-O-Pt structure in Pt ACs will not be reduced to metallic Pt during the HER process. Additionally, the previous publications related to Pt-based atomic clusters or single atoms also demonstrate that the Pt species can maintain their cationic nature after the HER test (*Nat. Energy* 2019, 4, 512-518; *J. Am. Chem. Soc.* 2020, 142, 5594-5601; *Energy Environ. Sci.* 2017, 10, 2450-2458; *Nat. Commun.* 2019, 10, 1743).

Fig. R5 The high resolution of Pt 4f of Pt-ACs/CoNC and after test at -0.5 V vs. RHE.

Comment 11: In the computational studies there is no indication of the Co center's interaction with something at the 6-position (e.g., from the first sublayer of the support). Justify this assumption.

Response 11: The Pt_7O_{12} cluster was constructed on a CoN_4 -carbon surface to simulate the atomic structure of Pt-ACs/CoNC. In this constructed atomic structure, only the Pt atoms (Pt1, Fig. 4a) are directly connected to Co atoms *via* the Pt-O-Co bonds having direct interactions. The Pt atoms at Pt2 and Pt3 positions do not indicate any direct interaction. This assumption is reasonably based on observation in characterization results. As shown in the HAADF-STEM images in Fig. 1c and d, the Co atoms are scattered around Pt clusters. Additionally, the Co atoms are existed as single atoms, indicating they are randomly dispersed on N-doped carbon substrate. In contrast, the Pt atomic clusters are distributed in ensembles that Pt atoms locate next to each other roughly on the same plane. If every Pt atom directly bonds with Co atoms, the distribution trend of Co atoms should be the same as Pt atoms (atomic cluster), which is contrary to the observation of Co atoms distributed as single atoms.

Additionally, we also would like to point out that it is not necessary for each Pt atom interacting with Co to have a HER active Pt site. Our analysis (from free energy studies on different Pt sites associated with Pt_7O_{12} , $\text{Pt}_{11}\text{O}_{16}$ and $\text{Pt}_{18}\text{O}_{24}$) shows that several Pt sites that found to be active are also far from the interface, i.e. they do not interact with Co. Thus, when considering Pt ACs structures, the interfacial electronic structure need not play a defining role in dictating HER, instead, the surface Pt sites on Pt ACs can be important in driving HER. The interface electronic structure, *via* the O atoms, plays an essential role in ensuring the Pt-O-Pt units bind to the CoN_4 units and does not lead to their aggregation during the preparation and reaction conditions.

Comment 12: In Fig 1j distributions are giving for C, N, Co and Pt, but not for O, although it is provided in the SI for the NP and SA catalysts. Also, no statistics are presented for correlations in the distributions (e.g., between Co and Pt and between O and Pt).

Response 12: The STEM-EDS elemental mapping of O element for Pt-ACs/CoNC is shown in Supplementary Fig. 6 as well as the HAADF-STEM image. Based on the reviewer's suggestion, we have added the correlations in the distributions of Co and Pt, O and Pt in Supplementary Fig 6 on Page 12.

Fig. R6 Elemental mapping of Co+Pt (a), Pt+O (b).

We have added the elemental mapping of Co+Pt and Pt+O in Supplementary Fig. 6c-d in the revised Supplementary Information.

Comment 13: *Are other CoNCs present besides the CoN4C pyridinic site? E.g., CoN3C or a pyrrolic sites?*

Are all N doped sites occupied by Co or might some N sites interact directly with Pt in the absence of Co (e.g., as in Muhich et al. 2013)? Pt-N or Pt-O

Response 13: Based on the characterization results in the manuscript, CoN4C pyridinic site should be the main atomic structure in CoNC. The EXAFS fitting results of CoNC in Supplementary Fig. 3 show the atomically dispersed Co atoms are stabilized by four N atoms in the equatorial plane and one O atom in the axial position, indicating the Co atoms are mainly stabilized by N atoms. Moreover, the high-resolution XPS spectrum of N 1s for CoNC in Supplementary Fig. 11a can be deconvoluted into three peaks at 398.64, 401.00, 403.00 eV, corresponding to pyridinic-N/metal-N, graphitic N, and oxidized N, respectively, indicating the absence or really low amount of pyrrolic N in CoNC.

Based on the XPS results, the atomic ratio of N is 4.48 at% and the atomic ratio of Co is 0.17 at% (weight ratio of Co is 3.5 wt% based on ICP-MS results). As each Co atom are bonded with four N atoms, which indicates not all N sites are occupied by Co during the preparation of CoNC. However, the EXAFS spectrum of Pt in Pt-ACs/CoNC (Fig. 2e) exhibits the only dominant peak locating around 1.65 Å, sharing a similar position to the typical exclusive Pt-O coordination environment in PtO₂, which demonstrates the Pt atoms in Pt-ACs/CoNC should be coordinated by O atoms instead of residue N atoms. The fitting results of Pt-ACs/CoNC (Fig. 2e) indicate the Pt atoms are stabilized by three Pt-O bonds in Pt-ACs/CoNC, further demonstrating the absence of Pt-N coordination in the Pt ACs.

Comment 14: *There is no explanation given for very similar rates in Figs 3a and c between Pt-O-ACs and traditional Pt on C catalysts.*

Response 14: We would like to thank the reviewer for the valuable comment. The data

shown in Fig 3a and 3c are obtained with the same total mass loading of $262 \mu\text{g cm}^{-2}$ for Pt-ACs/CoNC and commercial Pt/C (in which the element Pt loading is 1.31 and $52.4 \mu\text{g cm}^{-2}$ for Pt-ACs/CoNC and commercial Pt/C, respectively). Generally, commercial Pt/C is often considered a benchmark catalyst for HER, as commercial Pt/C usually exhibits excellent catalytic activity for HER with optimized mass loading (*Nat. Commun.*, 2014, 5, 3783). Our results show that both Pt-ACs/CoNC and commercial Pt/C display similar HER catalytic activity, suggesting both of them are excellent for hydrogen generation. However, when enlarge the LSV results in Fig. 3a in the Manuscript, the as-prepared Pt-ACs/CoNC exhibit slightly superior HER catalytic activity, achieving a j of 10 mA cm^{-2} at the overpotential of 24 mV (26 mV at 10 mA cm^{-2} for commercial Pt/C), demonstrating the excellent capability of Pt-ACs/CoNC for HER. Moreover, Pt-ACs/CoNC display much superior HER catalytic activity compared to commercial Pt/C with the same element Pt loading (Pt loading is $1.31 \mu\text{g cm}^{-2}$ for both samples), indicating the intrinsic catalytic activity of Pt ACs is superior compared to commercial Pt/C (Fig. R7).

Fig. R7 The electrochemical performance test of Pt-ACs/CoNC and commercial Pt/C with same Pt loading on the working electrode.

We have added the electrochemical performance test of Pt-ACs/CoNC and commercial Pt/C with same Pt loading on the working electrode (different total loading) in Supplementary Fig. 16c on Page 22 and the corresponding explanation in the revised Manuscript on Page 11.

Comment 15: *On line 310 the authors suggest that the different Tafel slopes suggest different HER pathways. However, the different kinetics could be explained by different activation barriers on the different active sites. DFT computed barriers could resolve this.*

Response 15: We would like to thank the reviewer for the valuable comment. Based on the reviewer's suggestions, we have explored the HER catalytic mechanism of Pt-ACs/CoNC by using DFT calculations (Volmer-Heyrovsky or Volmer-Tafel). In Volmer-Tafel pathway, two adsorbed hydrogen atoms on active sites combine to give H_2 . In contrast, in the Volmer-Heyrovsky pathway, the transfer of a second electron to the adsorbed hydrogen atom is coupled to the transfer of another proton from the

solution to evolve H_2 . We have modeled both these pathways using DFT and the free energy diagram is shown in Fig. R8. The results show that the Volmer-Tafel pathway is more favorable since the reaction energy toward H_2 evolution are lower. Thus, Volmer-Tafel pathway should be the dominant HER catalytic process of Pt-ACs/CoNC, which is consistent with our experimental results. On the other hand, we find a larger reaction energy for H_2 molecule to form at a Pt site of Pt-ACs/CoNC *via* the Volmer-Heyrovsky pathway, thus making it less probable as the possible reaction route. Therefore, both of the DFT calculations and experimental results demonstrate the Volmer-Tafel pathway is more favorable for Pt-ACs/CoNC in HER process.

Fig. R8 Free energy diagrams of the Volmer-Heyrovsky and Volmer-Tafel catalytic pathways on Pt-ACs/CoNC.

We have added free energy diagrams of the Volmer-Heyrovsky and Volmer-Tafel catalytic pathways on Pt-ACs/CoNC in Supplementary Fig. 24 on Page 30 in the revised Supplementary Information, and the explanations on Page 14 in the revised Manuscript and Page 30 in the revised Supplementary Information.

Page 30 in the revised Supplementary Information: We have explored the HER catalytic mechanism of Pt-ACs/CoNC by using DFT calculations (Volmer-Heyrovsky or Volmer-Tafel). In Volmer-Tafel pathway, two absorbed hydrogen atoms on active sites combine to give H_2 . In contrast, in the Volmer-Heyrovsky pathway, the transfer of a second electron to the absorbed hydrogen atom is coupled to the transfer of another proton from the solution to evolve H_2 . We have modeled both these pathways using DFT and the free energy diagram is shown in Supplementary Fig. 24. The results show that the Volmer-Tafel pathway is more favorable since the reaction energy toward H_2 evolution is lower. Thus, Volmer-Tafel pathway is the dominant HER catalytic process of Pt-ACs/CoNC, which is consistent with our experimental findings. Instead, we find that there is relatively larger reaction energy for H_2 molecule to form at a Pt site of Pt-ACs/CoNC via the Volmer-Heyrovsky pathway, thus making it less probable in our experiments. Therefore, both of the DFT calculations and experimental results demonstrate the Volmer-Tafel pathway is more favorable for Pt-ACs/CoNC in HER process.

Comment 16: *Why are no results reported for more reducing potentials than 256 mV?*

In Fig 3c the current density was still increasing with larger overpotential.

Response 16: The 256 mV refers to the overpotential for CoNC delivering a current density of 10 mA cm⁻². We have presented the current density of 40 mA cm⁻² for CoNC at the overpotential of 324 mV in Fig. 3a. Also as shown in Fig. 3a, we have presented the LSV results with the current density of 40 mA cm⁻² for all the catalysts with different overpotentials (324 mV for CoNC, 136 mV for Pt-SAs/CoNC, 56 mV for Pt-ACs/CoNC and 76 mV for Pt-NPs/CoNC), which is sufficient to compare the HER catalytic activity of the as-prepared catalysts.

Fig. 3c is the Tafel slope of CoNC, Pt-SAs/CoNC, Pt-ACs/CoNC and Pt-NPs/CoNC. The Tafel plot is useful for the determination of the reaction mechanism. Generally, the smaller the Tafel slope, the better the performance. Tafel slope was obtained by linear fitting the plot derived from the logarithm of current density vs. overpotential. Usually, the Tafel slope is calculated at lower overpotential because at large overpotential the mass transfer is the main control of the current. Therefore, for the Tafel slope calculation, we have chosen lower overpotential (negligible effect of mass transfer) to compare the reaction mechanism of each catalyst (*Nat. Energy*, 2019, 4, 512-518; *J. Am. Chem. Soc.* 2020, 142, 5594-5601; *Nat. Energy*, 2018, 3, 773-782, *Nat. Commun.*, 2019, 10, 4977).

Comment 17: *Structures in Figs 4b and c should not be rotated relative to 4a to make it easier to understand the density isosurface relative to the atomic structure.*

Response 17: Based on the reviewer's suggestion, we have made Fig. 4a with the same rotation angles with Fig. 4b and Fig. 4c to keep their consistency.

Reviewer #3:

The manuscript from Zhao et al. presents the preparation route, physicochemical characterization and the performance toward hydrogen evolution reaction (HER) of a class of promising ultra-low loading Pt catalyst. The excellent performance and stability toward HER of the presented material composed of ultrasmall Pt atomic clusters compared to single atom or nanoparticulate catalysts is rationalized by the stabilization of the Pt clusters by CoN₄ species from the carbon support, and synergetic electronic effects from Pt cluster composition and metal-support interactions.

To support their hypothesis, the authors used an impressive combination of advanced characterization techniques completed with DFT calculations. In overall the manuscript is clear and convincing, I would thus recommend publication in Nature Communications after the authors consider some minor revisions.

Response: We would like to thank the reviewer for the valuable comment. We have carefully revised the Manuscript and Supplementary Information according to the reviewer's comments. The revised content has been underlined and highlighted in yellow in the revised Manuscript and Supplementary Information. The point-by-point responses are shown in detail below.

Comment 1: *In the microscopy part of the manuscript, the X-EDS maps with linescan analyses are not all convincing. From Figures 1 and S2, the fact that N, Co or Pt elements appear also out of the carbon matrix suggests the signal/noise ratio (counting time) may not be sufficient. Whereas it can be enough to support the overall homogenous repartition of such elements at the ~100 nm scale, the linescans Figure 1.i and S5 on one single atomic cluster <1nm needs precautions.*

Can the authors show the X-EDS spectra restricted to the region of atomic cluster investigated and associated elemental map? If there is not enough resolution and/or intensity, there is no point in doing linescans.

Response 1: We would like to thank the reviewer for the valuable comment. The HAADF-STEM images of Pt-ACs/CoNC in Fig. 1i and Supplementary Fig. 5 are obtained at a low voltage of 60 kV. The line scan results regarding Pt ACs mainly show similar distribution trends of elements O and Pt over the clusters. We have also performed characterizations such as XPS, XAS and DFT to confirm the existence of Pt-O bonds (similar distribution trends of Pt and O) in Pt-ACs/CoNC, which could support the conclusion from the line scan.

Firstly, O 1s spectra of CoNC, Pt-SAs/CoNC, Pt-ACs/CoNC and Pt-NPs/CoNC in Supplementary Fig. 11b show the intensity of the peak belonging to O²⁻ increases with higher Pt loading (from SAs to NPs), whereas the intensity of the OH⁻ peak decreases correspondingly, attributing to the formed Pt-O bonds when Pt ions are incorporated. Secondly, the EXAFS results show the only dominant peak for Pt-ACs/CoNC locates around 1.65 Å, sharing a similar position to the typical exclusive Pt-O coordination environment in PtO₂, which demonstrates the Pt atoms in Pt-ACs/CoNC are coordinated by O atoms. Moreover, the EXAFS fitting results of Pt-ACs/CoNC (Fig. 2e) indicate the Pt atoms are stabilized by three Pt-O bonds in Pt-ACs/CoNC (inset of

Fig. 2e). Thirdly, the DFT calculations propose a Pt₇O₁₂ cluster constructed on CoN₄-carbon surface to simulate the atomic structure of Pt-ACs/CoNC (Fig. 4a). After DFT structural optimization, each Pt atom is stabilized by three O atoms and some of Pt atoms interact with Co SAs through O atom at the interface. This is consistent with the fitting results of EXAFS in Fig. 2e. All these results serve as direct evidence that the Pt atoms are stabilized by O atoms *via* the formation of Pt-O bonds in Pt-ACs/CoNC, thus the Pt and O atoms should display similar distribution trends. This is highly consistent with the line scan results.

We have followed the reviewer's suggestion and have tried to acquire X-EDS spectra restricted to the region of atomic cluster. However, it is really difficult to get the elemental map at such high magnifications. Slight drifting in the acquiring process (need more counting time) can significantly influence the quality of elemental mapping results. On the other hand, line scan is a much faster technique compared to elemental mapping, thus can endure such harsh requirement. Additionally, the other above-mentioned characterization results can also support our line scan results. Therefore, the results of line scans should be reliable.

Comment 2: *How do the authors extract sample height with single atom resolution from HAADF-STEM images in Figure 1h?*

Response 2: The sample height with single atom resolution from HAADF-STEM images has been extracted by using the Digital Micrograph software, which is the same method used in the reference of *ACS Catal.* 2019, 9, 5998-6005; *Nature. Commun.* 2021, 12, 2664.

Comment 3: *Please add some legend for the color code used in cartoons in Figures S14, 1f, 4.*

Response 3: Based on the reviewer's suggestion, we have added the color code for the atoms in the above Figures.

Comment 4: *When Figure S3 is mentioned in the text, the reader has no idea what is the model used behind the fit of the EXAFS curve. Maybe Figure S14 should be merged with Figure S3.*

Response 4: We would like to thank the reviewer for the valuable comment. The model in Supplementary Fig. 3 should be the atomic structure of CoNC, in which the isolated Co atoms are stabilized by four N atoms in the equatorial plane and one O atom in the axial position. The model in Supplementary Fig. 14 should be the atomic structure of Pt-SAs/CoNC with Pt single atoms stabilizing by three O atoms supported on CoNC substrate.

We have added the atomic model for CoNC in Supplementary Fig. 3 to better illustrate the atomic configurations of CoNC according to the EXAFS fitted data.

Comment 5: *Please replace 'spectra' per 'patterns' in figure S28 caption.*

Response 5: Following the reviewer's suggestion, we have changed 'spectra' to 'patterns' in Supplementary Fig. 28 caption (Supplementary Fig. 31 in the revised Supplementary Information).

'Supplementary Fig. 31 The XRD patterns of Ru-ACs/CoNC and Ir-ACs/CoNC.'

REVIEWERS' COMMENTS

Reviewer #1 (Remarks to the Author):

The authors have worked quite hard to revise the manuscript. The manuscript is now acceptable for publication.

Reviewer #2 (Remarks to the Author):

The authors have made an extensive effort to address the concerns raised by the reviewers and the manuscript is substantially better than it was in its original form.

Although this is only mentioned in the response and not in the manuscript, I strongly disagree with the response by the reviewers (Reviewer 2, Comment 8, Page 21) that the effect of applied bias can be taken into account by the computational hydrogen electrode and that it is a linear effect. Our own work has shown that this is not the case including published articles (see <https://doi.org/10.1021/jacs.1c03428>, and <https://doi.org/10.1021/acs.jpcc.1c07484>), and several works in progress. Fortunately, the authors do not make this statement in the manuscript, but the CHE is highly misleading as it is becoming apparent that most reaction energies depend non-linearly on bias and I hope the authors refrain from making this statement in the future.

The ms is now suitable for publication.

Reviewer #3 (Remarks to the Author):

The gave appropriate answers to this reviewers' comments, who recommend publication.

Detailed Reviewer Response

Reviewer #1:

The authors have worked quite hard to revise the manuscript. The manuscript is now acceptable for publication.

Response: We appreciate the reviewer's recommendation very much.

Reviewer #2:

The authors have made an extensive effort to address the concerns raised by the reviewers and the manuscript is substantially better than it was in its original form.

Although this is only mentioned in the response and not in the manuscript, I strongly disagree with the response by the reviewers (Reviewer 2, Comment 8, Page 21) that the effect of applied bias can be taken into account by the computational hydrogen electrode and that it is a linear effect. Our own work has shown that this is not the case including published articles (see <https://doi.org/10.1021/jacs.1c03428>, and <https://doi.org/10.1021/acs.jpcc.1c07484>), and several works in progress. Fortunately, the authors do not make this statement in the manuscript, but the CHE is highly misleading as it is becoming apparent that most reaction energies depend non-linearly on bias and I hope the authors refrain from making this statement in the future. The ms is now suitable for publication.

Response: We appreciate the reviewer's recommendation very much.

We thank the reviewer for pointing us to the relevant references. We have gone through the papers and indeed acknowledge that the CHE method is limited. We also see that to use the GC approach, we would need the JDFTx code or something similar, which at present is beyond the scope of this work. We shall look into this novel method in our future works.

Reviewer #3:

The gave appropriate answers to this reviewers' comments, who recommend publication.

Response: We appreciate the reviewer's recommendation very much.